# Intermittent fasting promotes type 3 innate lymphoid cells secreting IL-22 contributing to the beigeing of white adipose tissue

Hong Chen[1,2†], Lijun Sun[1†], Lu Feng[1], Xue Han[1], Yunhua Zhang[1], Wenbo Zhai[1], Zehe Zhang[1], Michael Mulholland[3], Weizhen Zhang[1,3]*, Yue Yin[4]*

[1]Department of Physiology and Pathophysiology, School of Basic Medical Sciences, Center for Reproductive Medicine, Third Hospital, Peking University, Beijing, China; [2]State Key Laboratory of Female Fertility Promote, Center for Reproductive Medicine, Department of Obstetrics and Gynecology, Peking University Third Hospital, Beijing, China; [3]Department of Surgery, University of Michigan Medical Center, Ann Arbor, United States; [4]Department of Pharmacology, School of Basic Medical Sciences, and Key Laboratory of Molecular Cardiovascular Science, Ministry of Education, Peking University, Beijing, China

**\*For correspondence:**
weizhenzhang@bjmu.edu.cn
(WZ);
yueyin@bjmu.edu.cn (YY)

†These authors contributed
equally to this work

**Competing interest:** The authors
declare that no competing
interests exist.

**Reviewing Editor:** Kiyoshi
Takeda, Osaka University, Japan

**Abstract** Mechanism underlying the metabolic benefit of intermittent fasting remains largely unknown. Here, we reported that intermittent fasting promoted interleukin-22 (IL-22) production by type 3 innate lymphoid cells (ILC3s) and subsequent beigeing of subcutaneous white adipose tissue. Adoptive transfer of intestinal ILC3s increased beigeing of white adipose tissue in diet-induced-obese mice. Exogenous IL-22 significantly increased the beigeing of subcutaneous white adipose tissue. Deficiency of IL-22 receptor (IL-22R) attenuated the beigeing induced by intermittent fasting. Single-cell sequencing of sorted intestinal immune cells revealed that intermittent fasting increased aryl hydrocarbon receptor signaling in ILC3s. Analysis of cell-cell ligand receptor interactions indicated that intermittent fasting may stimulate the interaction of ILC3s with dendritic cells and macrophages. These results establish the role of intestinal ILC3s in beigeing of white adipose tissue, suggesting that ILC3/IL-22/IL-22R axis contributes to the metabolic benefit of intermittent fasting.

## eLife assessment

This study provides **valuable** findings showing the production of IL-22 from intestinal ILC3 during intermittent fasting promotes beigeing of white adipose tissue. The authors provided **solid** data and mechanistic insight by which IL-22-derived from ILC3 directly induces beigeing.

## Introduction

Obesity is defined as an epidemic metabolic disease characterized by excessive fat accumulation as the consequence of long-term energy surplus. Thus, therapeutic management of obesity has been focused on the restoration of energy balance, either by decreasing energy intake or by increasing energy expenditure (*Liu et al., 2021*). Among these strategies, intermittent fasting is becoming a popular dietary approach. All intermittent fasting schemes including alternate day fasting, 5:2 intermittent fasting, and daily time-restricted feeding have shown the health benefits such as delaying aging (*Colman et al., 2009*; *Mattison et al., 2012*; *Mattison et al., 2017*; *Ulgherait et al., 2021*;

**eLife digest** Obesity refers to a condition where a person has excessive fat accumulation, which can have negative impacts on their health. Managing obesity has typically relied on reducing energy intake and increasing energy use through diets and exercise.

For example, intermittent fasting is a diet strategy involving periods of time in a day or week where a person does not eat any food. Research has shown that intermittent fasting may improve the metabolism and increase energy use by enhancing a process known as "beigeing" of white fat tissue.

In this process, white fat cells or their precursor cells differentiate into beige fat cells, which can consume excess energy by burning fat. Consequently, understanding how beigeing of white fat cells is activated in intermittent fasting may reveal a promising strategy for tackling obesity and metabolic diseases.

Immune cells found in the gut known as innate lymphoid cells (ILCs) may play a role in the metabolic benefits from intermittent fasting. However, the roles of ILCs are complex: some types of ILCs can promote obesity, while others show metabolic benefits through their release of proteins like IL-17 and IL-22, which can help the body to metabolise glucose.

To find out if these immune cells play a role in intermittent fasting, Chen, Sun et al. used diet-induced obese mice that had to fast every other day. Intermittent fasting was found to cause a form of ILCs (ILC3s) to release IL-22, which resulted in beigeing of white fat cells in obese mice. Single-cell sequencing techniques of gut immune cells further revealed that intermittent fasting increased forms of signalling in ILC3s and caused ILC3s to interact with other immune cells, such as dendritic cells and macrophages.

The findings demonstrate how intermittent fasting causes beigeing of white adipose tissue through ILC3s, revealing mechanisms underpinning the metabolic benefits found from intermittent fasting. More research into this process may help identify new targets for treating obesity.

*Stekovic et al., 2019*), improving metabolism (*de Cabo and Mattson, 2019*; *Hepler et al., 2022*), and enhancing cognition (*Mattson et al., 2018*; *Liu et al., 2019*; *Mattson and Arumugam, 2018*). The mechanism underlying metabolic benefit of intermittent fasting remains largely unknown. Its metabolic benefit was initially attributed to limitation of energy intake. Recent studies have indicated an alternative mechanism involving beigeing of white adipose tissue, which accounts for the major plasticity of energy expenditure. Studies by *Kim et al., 2017*, have shown that intermittent fasting induces white adipose beigeing via stimulation of angiogenesis and macrophage M2 polarization. On the other hand, studies by *Li et al., 2017*, have indicated that alternate day fasting induces white adipose tissue beigeing by shaping the gut microbiota and elevating the fermentation products acetate and lactate. Although gut microbiota is closely related to immune response, it remains unclear whether intestinal immune cells contribute to the metabolic benefit of intermittent fasting.

Innate lymphoid cells (ILCs) are a group of natural immune cells lacking antigen-specific receptors expressed on T cells and B cells (*Gasteiger et al., 2015*). Based on developmental trajectories, ILCs are divided into five groups: natural killer cells, type 1 ILCs, type 2 ILCs, type 3 ILCs (ILC3s), and lymphoid tissue-inducing cells (*Vivier et al., 2018*). Adipose-resident type 1 ILCs can promote adipose tissue fibrosis and obesity-associated insulin resistance (*O'Sullivan et al., 2016*; *Wang et al., 2019*) while type 2 ILCs promote beiging of white adipose tissue and limit obesity (*Brestoff et al., 2015*; *Lee et al., 2015*; *Wang et al., 2021b*). However, the contribution of ILC3s on adipocytes and obesity are less clear. ILC3s can produce interleukin-17 (IL-17) and interleukin-22 (IL-22) in response to extracellular bacteria and fungi (*Sanos et al., 2009*). ILC3s-derived IL-22 can enhance the intestinal mucosal barrier function, reduce endotoxemia and inflammation, ameliorate insulin sensitivity (*Wang et al., 2014*; *Hasnain et al., 2014*), and improve the metabolic disorder of polycystic ovary syndrome (*Qi et al., 2019*). However, ILC3s are also reported to be involved in the induction of obesity (*Sasaki et al., 2019*), contributing to the metabolic disease (*Sasaki et al., 2019*; *Upadhyay et al., 2012*; *Wang et al., 2017*; *Kawano et al., 2022*). Therefore, the role of ILC3s in metabolic disease seems complex and the role of ILC3s in intermittent fasting and beigeing of adipose tissue is not known.

Here, we showed that alternate day fasting promoted the secretion of IL-22 by ILC3s. Further, adoptive transfer of intestinal ILC3s increased thermogenesis in diet-induced obesity (DIO) mice.

Exogenous IL-22 induced the beigeing of white adipose tissue. Deficiency of IL-22 receptor attenuated the beigeing of white adipose tissue induced by intermittent fasting. Our study demonstrates that intestinal ILC3-IL-22-IL-22R axis is actively involved in the regulation of adipose tissue beigeing. Our findings thus reveal a novel pathway in the dialog between the gut and adipose tissue.

## Results

### Intermittent fasting enhances IL-22 production by intestinal ILC3s

To explore the effect of intermittent fasting on intestinal immune cells, we applied alternate day fasting to mice fed normal chow diet (NCD-IF group) or high-fat diet (HFD-IF group) (*Figure 1A*). Intermittent fasting significantly reduced the body weight of mice fed HFD, while demonstrating no effect on food intake rate which was normalized to body weight (*Figure 1—figure supplement 1A and B*). Moreover, intermittent fasting decreased the respiratory quotient (RQ) on the fasting day while increased the energy consumption on the feeding day in mice fed HFD (*Figure 1—figure supplement 1C and D*), improved glucose and lipid metabolism in mice fed HFD (*Figure 1—figure supplement 1E and F*), and promoted white adipose tissue beigeing in mice fed NCD or HFD (*Figure 1—figure supplement 2*).

To explore whether gut immune system contributes to the effects of intermittent fasting on white adipose tissue beigeing, we examined levels of various cytokines in intestine. Notably, mRNA level of *Il22* in the NCD-IF group was significantly higher relevant to the control group (*Figure 1B*). Consistently, plasma concentration of IL-22 detected by enzyme-linked immunosorbent assay (ELISA) was also increased (*Figure 1C*). These results suggest that intermittent fasting increases levels of IL-22 in the intestine and plasma. Since ILC3s are the main source of IL-22 in the small intestine (*Seillet et al., 2020*; *Victor et al., 2017*; *Gronke et al., 2019*), we detected the proportion of ILC3s in the lamina propria of small intestine using flow cytometry. In mice fed NCD, intermittent fasting significantly increased proportion of IL-22 positive ILC3s in the small intestine lamina propria (siLP) (*Figure 1D*), while demonstrating no effect on the percentile of total ILC3s (*Figure 1E*). Interestingly, intermittent fasting did not influence the secretion of IL-22 of T cells marked as lineage⁺ Rorγt⁺ cells (*Figure 1F*). Besides, intermittent fasting didn't alter the levels of ILC3s and IL-22 in mouse adipose tissue (*Figure 1G* and *Figure 1—figure supplement 3*). Similar to NCD mice, intermittent fasting significantly increased mRNA levels of IL-22 in the intestine of HFD mice while didn't influence the secretion of IL-22 by T cells (*Figure 1H and I*). Furthermore, mice fed HFD showed an obvious reduction in the plasma IL-22 (*Figure 1J*) and percentage of total and IL-22 positive ILC3s (*Figure 1K*). Further, the decrement of plasma IL-22 (*Figure 1J*) as well as IL-22 positive ILCs in intestine (*Figure 1K*) was attenuated by 30 days' intermittent fasting.

In order to explain the chronological relationship between ILC3s secreting IL-22 and beigeing of white adipose tissue, the mice were exposed to one cycle of intermittent fasting for 2 days. At this time, the body weight of mice didn't change and beige adipocytes haven't been induced (*Figure 1—figure supplement 4A and B*). However, significant increase in proportion of IL-22 positive ILC3s was induced by intermittent fasting for 2 days whereas total percentile of ILC3s remained unaltered (*Figure 1—figure supplement 4C*). These results indicate that the beigeing of white adipose tissue are subsequent to its effect on ILC3s secreting IL-22.

### Intestinal ILC3s promote beigeing of white adipose tissue

Next, we examined whether adoptive transfer of intestinal ILC3s can increase beigeing of white adipose tissue in DIO mice. Intestinal ILC3s were isolated and purified from the siLP of NCD mice (*Figure 2—figure supplement 1A*). These cells were defined as lineage⁻CD127⁺KLRG1⁻c-Kit⁺ cells (*Figure 2—figure supplement 1B*). As shown in *Figure 2A*, DIO mice transferred with ILC3s demonstrated a significant increment in the proportion of small intestinal ILC3s. Previous researches report that intestinal ILC3s specifically express gut homing receptors CCR7, CCR9, and α4β7 (*Kim et al., 2015*; *Mackley et al., 2015*; *Yu et al., 2021*), which may explain transplantation of intestinal ILC3s can migrate mainly to the intestine instead of adipose tissue. Plasma concentration of IL-22 also increased significantly relevant to phosphate-buffered saline (PBS) control (*Figure 2B*). Relevant to PBS control, adoptive transfer of intestinal ILC3s decreased body weight and weight of sWAT (subcutaneous white adipose tissue) slightly, while had no impact on food intake and the liver weight

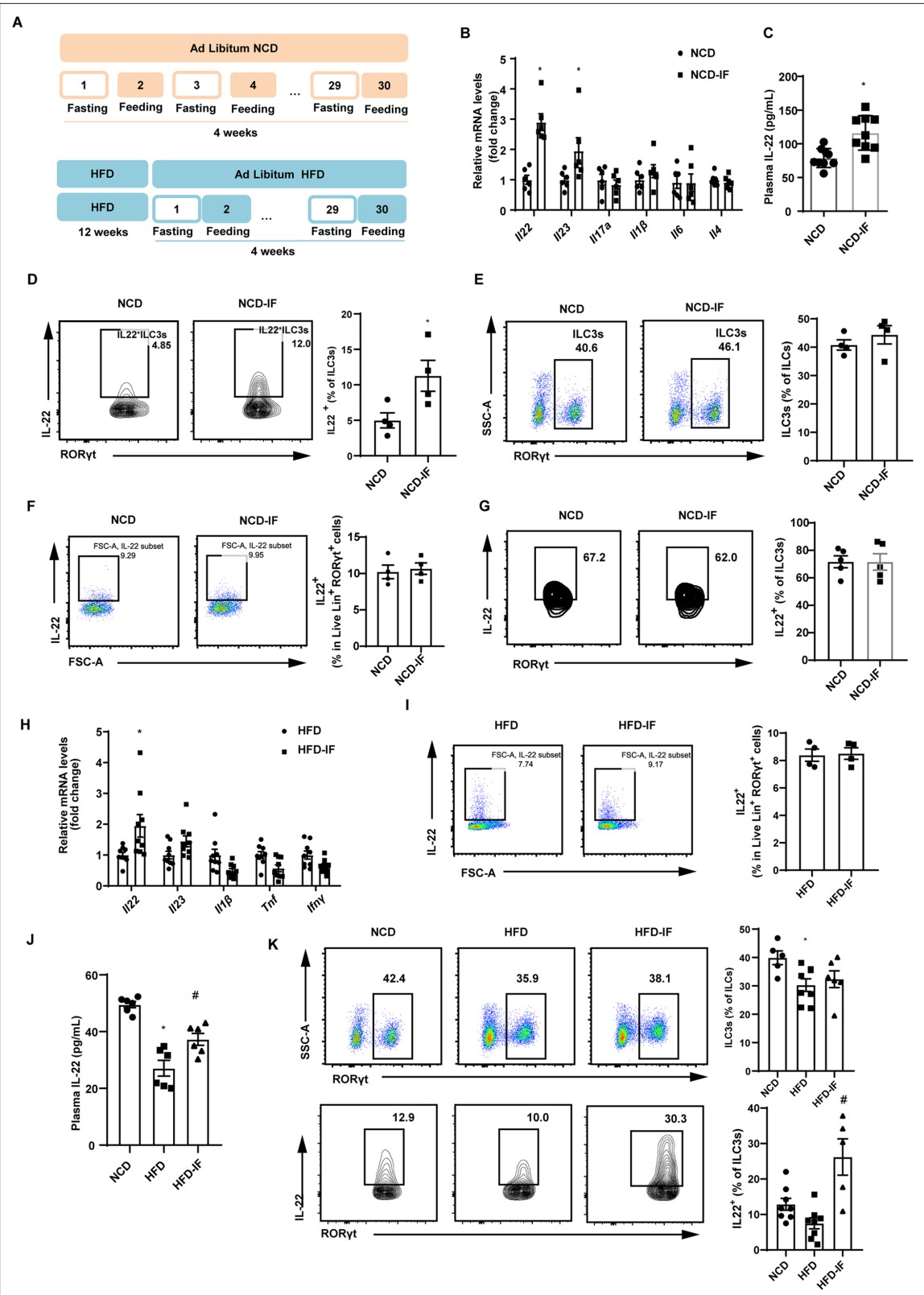

**Figure 1.** Intermittent fasting enhances interleukin-22 (IL-22) production by intestinal type 3 innate lymphoid cells (ILC3s). (**A**) Schematic illustration of the alternate day fasting regimen. NCD, normal chow diet. HFD, high-fat diet. We applied alternate day fasting to mice fed normal chow diet (NCD-IF group) or high-fat diet (HFD-IF group). The control groups were at free access to NCD (NCD group) or HFD (HFD group). n=9 for each group. (**B**) mRNA expression levels of cytokine genes in the small intestine of NCD and NCD intermittent fasting (NCD-IF) mice. qPCR results were normalized to β-actin.

*Figure 1 continued on next page*

*Figure 1 continued*

n=6 for each group. (**C**) Protein levels of IL-22 in plasma of NCD and NCD-IF mice. n=9 for each group. (**D**) Fl IL-22$^+$ cells in live CD127$^+$ lineage$^-$ RORγt$^+$ ILC3s from the small intestine lamina propria (siLP) of NCD and NCD-IF mice. Four independent experiments were performed with similar results. n=4. (**E**) Flow cytometric analysis of RORγt$^+$ ILC3s in live CD127$^+$ lineage$^-$ ILCs in the siLP of NCD and NCD-IF mice. n=4. (**F**) Flow cytometric analysis of IL-22$^+$ cells in live lineage$^+$ RORγt$^+$ cells in the siLP of mice fed NCD with or without intermittent fasting. n=4. (**G**) Flow cytometric analysis of IL-22$^+$ cells in CD90.2$^+$ lineage$^-$ RORγt$^+$ ILC3s from the stromal vascular fraction (SVF) cells of subcutaneous white adipose tissue (sWAT) in mice fed NCD with or without intermittent fasting. n=5. (**H**) mRNA expression levels of cytokine genes in the small intestine of mice fed HFD with or without intermittent fasting. qPCR results were normalized to β-actin. n=9. (**I**) Flow cytometric analysis of IL-22$^+$ cells in live lineage$^+$ RORγt$^+$ cells in the siLP of mice fed HFD with or without intermittent fasting. n=4. (**J**) Levels of IL-22 in plasma of NCD mice, HFD mice, and HFD-IF mice. n=6. (**K**) Flow cytometric analysis of RORγt$^+$ ILC3s in live CD127$^+$ lineage$^-$ ILCs and flow cytometric analysis of IL-22$^+$ cells in live CD127$^+$ lineage$^-$ RORγt$^+$ ILC3s from the siLP of NCD mice, HFD mice, and HFD-IF mice. n=5–8. * vs NCD, # vs HFD, p<0.05. All data represent the mean ± s.e.m. Statistical significance was determined by unpaired two-tailed Student's t test (**A–I**) or one-way ANOVA (**J** and **K**). NCD, normal chow diet. HFD, high-fat diet. NCD-IF, normal chow diet with intermittent fasting. HFD-IF, high-fat diet with intermittent fasting.

The online version of this article includes the following source data and figure supplement(s) for figure 1:

**Figure supplement 1.** Intermittent fasting improves glucose metabolism and lipid metabolism.

**Figure supplement 2.** Intermittent fasting promotes white adipose tissue beigeing.

**Figure supplement 2—source data 1.** Original file for the western blot analysis in *Figure 1—figure supplement 2E* (anti-UCP1, anti-β-actin).

**Figure supplement 2—source data 2.** PDF containing original scans of the relevant western blot analysis (anti-UCP1, anti-β-actin) with highlighted bands and sample labels.

**Figure supplement 3.** Intermittent fasting demonstrates no effect on the number of type 3 innate lymphoid cells (ILC3s) in subcutaneous white adipose tissue (sWAT).

**Figure supplement 4.** Short-term intermittent fasting induces intestinal type 3 innate lymphoid cells (ILC3s) to secrete interleukin-22 (IL-22).

(*Figure 2—figure supplement 1C–E*). In addition, ILC3s from CD45.1 mouse intestinal lamina propria lymphocytes were adoptively transferred into recipient mice, and CD45.1 positive immune cells were significantly increased in intestine but not in adipose tissue in mice transferred with ILC3s (*Figure 2—figure supplement 1F*), indicating the feasibility of ILC3s adoptive transfer. Notably, DIO mice transferred with intestinal ILC3s showed improved glucose tolerance (*Figure 2C*) and decreased levels of random blood glucose (*Figure 2D*). Cold exposure experiment showed that DIO mice transferred with intestinal ILC3s maintained significantly higher rectal temperatures during a 6 hr cold challenge (*Figure 2E*). Consistently, key thermogenic genes in sWAT including *Ucp1* and *Cidea* were significantly increased (*Figure 2F*). Size of adipocytes in the sWAT and eWAT was markedly reduced (*Figure 2G and H*). Furthermore, differentiated stromal vascular fraction (SVF) cells co-cultured with intestinal ILC3s in vitro (*Figure 2I*) demonstrated a significant increment in the expression of *Ucp1* and *Pparg* (*Figure 2J*), as well as a concurrent decrement in the size of lipid droplets (*Figure 2K*). On the other hand, co-culture with CD127$^-$ cells demonstrated no effect (*Figure 2J and K*). The in vitro experiment indicates the direct effect of ILC3s on adipocytes.

## Exogenous IL-22 increases beigeing of white adipose tissue

The role of type 3 immunity in DIO and metabolic syndrome is complex. ILC3-derived IL-22 are beneficial in metabolic syndrome (*Wang et al., 2014*; *Zou et al., 2018*) but can also contribute to metabolic disease (*Sasaki et al., 2019*; *Upadhyay et al., 2012*; *Wang et al., 2017*). In order to examine the role of promoting beigeing of white adipose tissue of IL-22 in the context of NCD and HFD mice, we intraperitoneally administrated IL-22 at the dose of 4µg/kg body weight/every other day for 6weeks into mice fed NCD or HFD. Saline was used as control. Administration of IL-22 increased its plasma concentration in mice fed either NCD or HFD (*Figure 3—figure supplement 1A*). Exogenous IL-22 significantly increased oxygen consumption, carbon dioxide production, and energy expenditure in mice fed either NCD or HFD (*Figure 3A–F*). RQ was significantly reduced in mice fed NCD at dark and mice fed HFD at light (*Figure 3G and H*). No significant difference was observed for animal activity (*Figure 3—figure supplement 1*). In addition, exogenous IL-22 significantly improved glucose tolerance in mice fed HFD (*Figure 4A*). Interestingly, body weight and food intake were not altered (*Figure 3—figure supplement 1*), indicating that the metabolic benefit of IL-22 is not dependent on food intake. Consistent with previous research, IL-22 administration improves insulin sensitivity without change in body weight (*Qi et al., 2019*). In addition, IL-22 can increase Akt phosphorylation in muscle, liver, and adipose tissues, leading to improvement in insulin sensitivity (*Wang et al., 2014*).

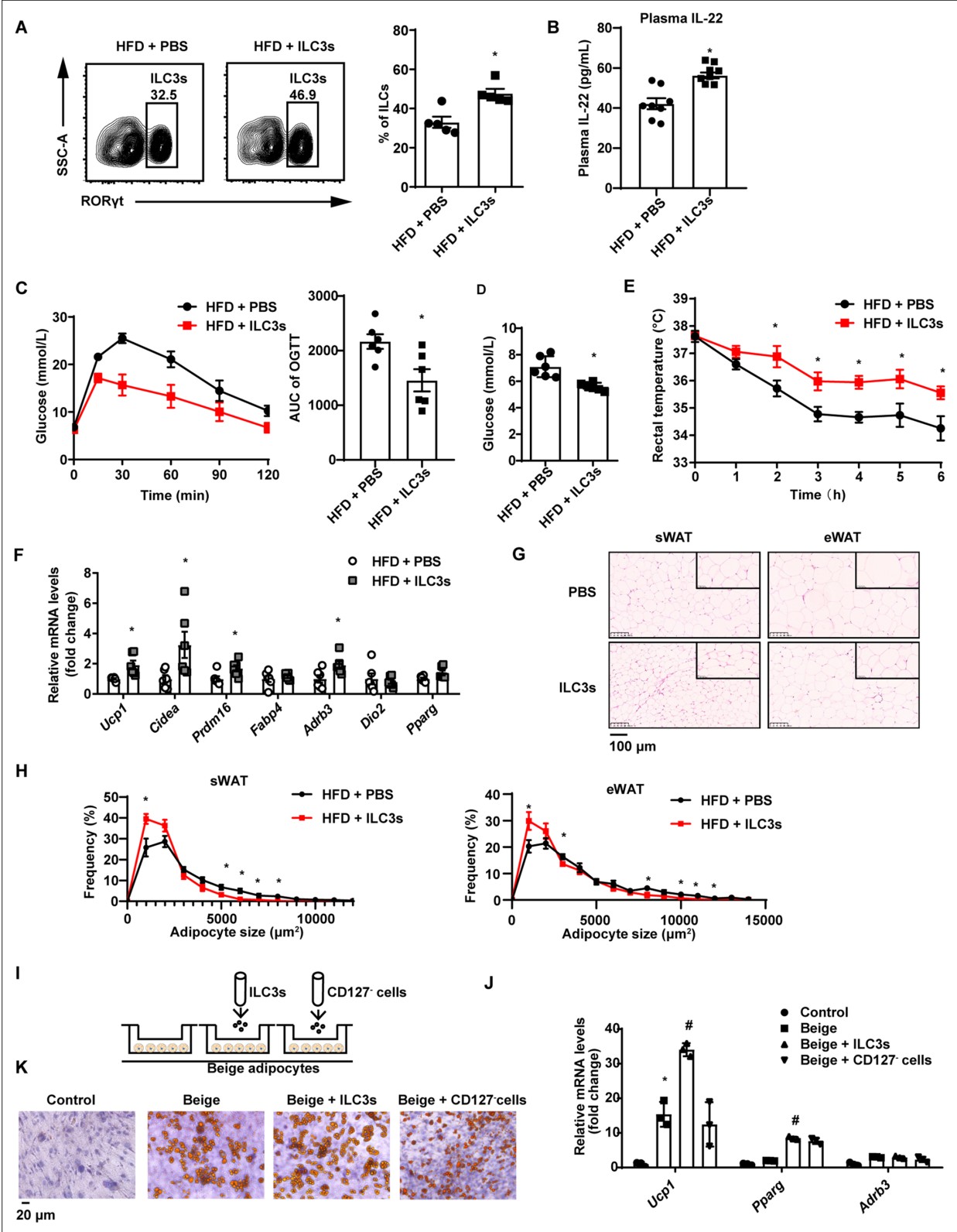

**Figure 2.** Type 3 innate lymphoid cells (ILC3s) promote beigeing of white adipose tissue. Six-week-old male C57BL/6J SPF wild-type mice were fed with high-fat diet (HFD) for 16weeks and then injected with ILC3s (HFD + ILC3s group) or phosphate-buffered saline (PBS) (HFD+PBS group) intravenously six times in a month. n=8 for each group. (**A**) Flow cytometric analysis of RORγt[+] ILC3s in live CD127[+] lineage[-] ILCs from the small intestine lamina propria (siLP) of mice transferred with PBS or ILC3s. The proportion of ILC3s in ILCs was shown in the histogram. n=5 for each group. (**B**) Levels of interleukin-22

*Figure 2 continued on next page*

*Figure 2 continued*

(IL-22) in plasma. n=8 for each group. (**C**) Oral glucose tolerance test (OGTT) and area under the curve (AUC). n=6mice/group. (**D**) Random blood glucose. n=6mice/group. (**E**) Rectal temperature of HFD mice transferred with PBS or ILC3s during a 6hr cold challenge (4°C). n=6mice/group. (**F**) qPCR analysis of thermogenic genes in subcutaneous white adipose tissue (sWAT). n=6mice/group. (**G**) Representative images of hematoxylin-and-eosin-stained sections of sWAT, epididymal white adipose tissue (eWAT), and BAT from HFD mice transferred with ILC3s or control (n=5). (**H**) Distribution and average adipocyte size of sWAT and eWAT were shown (n=5). (**I**) Schematic depicting the co-culture of ILC3s with stromal vascular fraction (SVF)-derived beige adipocytes. (**J**) qPCR analysis of thermogenic genes in SVF-derived cells. * indicates p<0.05vs control, # denotes p<0.05vs beige. n=3. (**K**) Oil red O staining of adipocytes after co-culture with ILC3s or CD127⁻ cells. All data represent the means ± s.e.m. Statistical significance was determined by unpaired two-tailed Student's t test (**A–H**) or one-way ANOVA (J).

The online version of this article includes the following figure supplement(s) for figure 2:

**Figure supplement 1.** Adoptive transfer of type 3 innate lymphoid cells (ILC3s) has no effect on the body weight in mice fed high-fat diet (HFD).

We next explored the effect of exogenous IL-22 on thermogenesis induced by 4°C cold exposure. Exogenous IL-22 rendered mice fed HFD resistant to core temperature drop induced by cold exposure (*Figure 4B*). Relevant to the pale yellow color in the control animals, subcutaneous fat of obese mice treated with exogenous IL-22 appeared dark yellowish (*Figure 4C*). The mRNA levels of thermogenic genes such as *Ucp1* were significantly increased by IL-22 (*Figure 4D*). Adipocyte size of subcutaneous adipose tissue and epididymal adipose tissue was significantly reduced in the animals treated with IL-22 (*Figure 4E and F*). These observations suggest that intraperitoneal injection of IL-22 promotes the beigeing of white adipose tissue in mice.

To determine whether IL-22 can directly act on adipocytes to promote their beigeing, adipose tissue SVF was isolated and induced for beige differentiation. Il22ra1 mainly expressed on mature adipocytes (*Figure 4—figure supplement 1A*). IL-22 at the dose of 100 ng/mL was continuously administered during beige differentiation. IL-22 significantly increased SVF beige differentiation evidenced by cell morphology, increment in mRNA levels of genes relevant to thermogenesis, including *Ucp1* and *Cidea*, and protein level of UCP1 (*Figure 4G–I*). IL-22 did not alter the levels of genes related to adipogenesis (*Figure 4—figure supplement 1B*). As expected, IL-22 increased the phosphorylation of STAT3 and MAPK (*Figure 4I*, *Figure 4—figure supplement 1C*), which can increase the expression of thermogenic genes (*Li et al., 2021*; *Zhang et al., 2016*). These results suggest that IL-22 can directly stimulate beigeing of white adipose tissue.

## IL-22RKO blocks beigeing induced by intermittent fasting

To explore whether IL-22R mediates the effect of intermittent fasting on the beigeing of white adipose tissue, IL-22R knockout (IL-22RKO) mice and wild-type (WT) littermates were subjected to alternate day fasting diet for 30 days. Rectal temperature of WT-IF and IL-22RKO-IF mice was monitored during 2 consecutive days of intermittent fasting. On the fasting day, the rectal temperature of IL-22RKO-IF mice was significantly lower than that of WT-IF mice at 16:00 time point (*Figure 5A*). On the feeding day, the rectal temperature of IL-22RKO-IF mice was lower at three time points: 8:00, 12:00, and 20:00 (*Figure 5B*). These results indicate that IL-22RKO attenuates the thermogenesis induced by intermittent fasting. Knockout of IL-22R demonstrated no effect on the glucose tolerance and insulin sensitivity in mice fed NCD (*Figure 5C and D*). However, the weight of subcutaneous adipose tissue in IL-22RKO-IF mice increased significantly (*Figure 5E*). mRNA levels of the thermogenic gene *Ucp1* decreased substantially, whereas *Fabp4* increased (*Figure 5F*). Furthermore, knockout of IL-22R significantly attenuated the increment of multilocular lipid droplets and the decrement of adipocyte size in the subcutaneous fat induced by intermittent fasting (*Figure 5G and H*), indicating a reduction in beigeing.

To further explore whether IL-22R mediates the effect of ILC3s on beigeing of adipocytes, intestinal ILC3s isolated from intermittent fasting mice were co-cultured with beige adipocytes from subcutaneous SVF of WT or IL-22RKO mice (*Figure 5I and J*). Co-culture with ILC3s significantly increased the mRNA levels of thermogenic genes in beige adipocytes derived from WT mice, while demonstrating no effect on beige adipocytes derived from IL-22RKO mice (*Figure 5K*). These observations indicate that IL-22R mediates the effect of intestinal ILC3s on beigeing of white adipocytes.

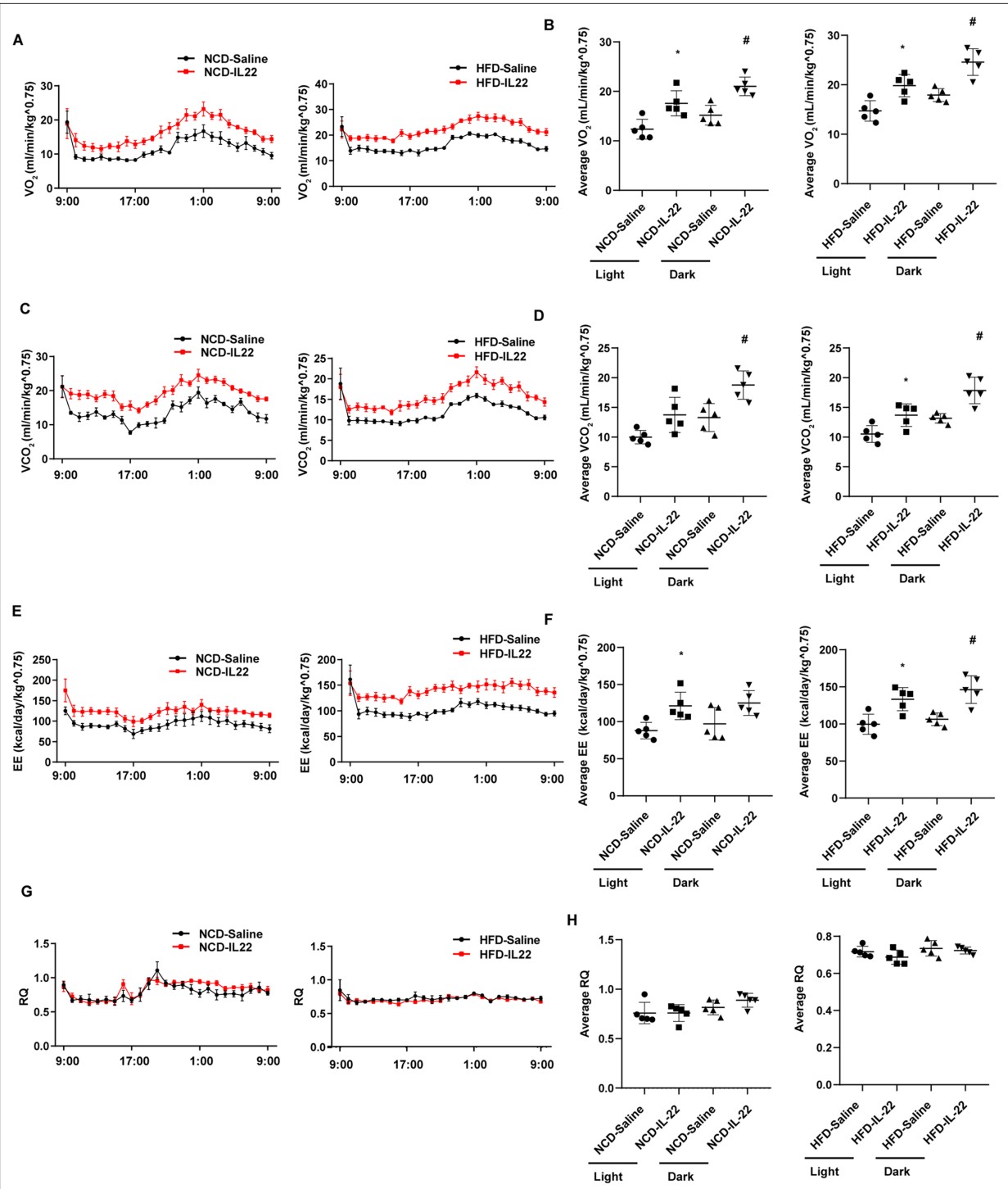

**Figure 3.** Interleukin-22 (IL-22) promotes energy expenditure. Six-week-old male C57BL/6J SPF wild-type mice were fed normal chow diet (NCD) or high-fat diet (HFD) for 12 weeks and then divided into four groups (NCD-saline, NCD-IL-22, HFD-saline, HFD-IL-22). Mice were intraperitoneally injected with 4 μg/kg IL-22 every other day for 6 weeks. The control groups were injected with saline. (**A**) $VO_2$ of mice fed NCD or HFD. (**B**) Average $VO_2$ at light and dark respectively. (**C**) $VCO_2$ of mice fed NCD or HFD. (**D**) Average $VCO_2$ at light and dark respectively. (**E**) Energy expenditure of mice fed NCD or HFD. (**F**) Average energy expenditure at light and dark respectively. (**G**) Respiratory quotient (RQ) of mice fed NCD or HFD. (**H**) Average RQ at light and dark respectively. * indicates $p<0.05$ vs NCD-saline or HFD-saline at light. # indicates $p<0.05$ vs NCD-saline or HFD-saline at dark. n = 5. Statistical significance was determined by one-way ANOVA (**B, D, F, H**).

The online version of this article includes the following figure supplement(s) for figure 3:

**Figure supplement 1.** Exogenous interleukin-22 (IL-22) has no effect on the body weight of mice.

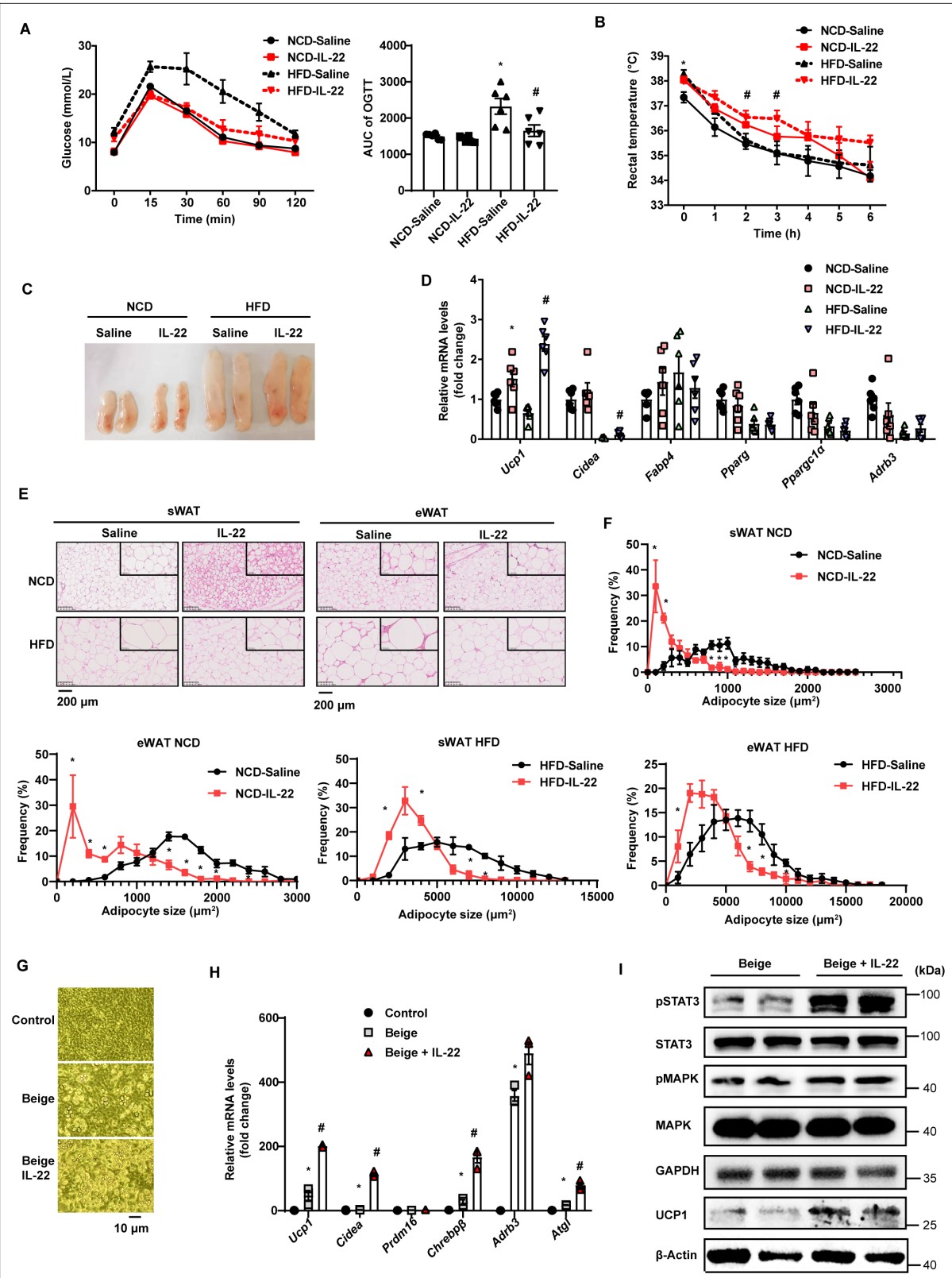

**Figure 4.** Interleukin-22 (IL-22) promotes beigeing of white adipose tissue. Six-week-old male C57BL/6J SPF wild-type mice were fed normal chow diet (NCD) or high-fat diet (HFD) for 12weeks and then divided into four groups (NCD-saline, NCD-IL-22, HFD-saline, HFD-IL-22). Mice were intraperitoneally injected with 4μg/kg IL-22 every other day for 6weeks. The saline group was injected with saline. n=6mice/group. (**A**) Oral glucose tolerance test (OGTT) and area under the curve (AUC). n=6mice/group. * indicates p<0.05vs NCD-saline; # denotes p<0.05vs HFD-saline. (**B**) Rectal temperature of mice

*Figure 4 continued on next page*

**Figure 4 continued**

during a 6hr cold challenge (4°C). n=6. * indicates p<0.05vs NCD-saline; # denotes p<0.05vs HFD-saline. (**C**) Representative image of sWAT of the four groups, NCD-saline, NCD-IL-22, HFD-saline, HFD-IL-22. (**D**) qPCR analysis of thermogenic genes of sWAT. n=6mice/group. * indicates p<0.05vs NCD-saline; # denotes p<0.05vs HFD-saline. (**E**) Representative images of hematoxylin-and-eosin-stained sections of sWAT and eWAT (n=5 for each group). (**F**) The distribution and average adipocyte size of sWAT and eWAT were determined by ImageJ. (**G**) Phase-contrast microscopic images of stromal vascular fraction (SVF) cells and adipocytes. (**H**) qPCR analysis of thermogenic genes in SVF cells and beige adipocytes. n=3. # denotes p<0.05vs beige. (**I**) pSTAT3, STAT3, pMAPK, MAPK, GAPDH, UCP1, β-actin protein expression in the beige adipocytes or beige adipocytes treated with IL-22 detected by western blotting. GAPDH and β-actin was used as the loading control. All data represent the mean ± s.e.m. Statistical significance was determined by one-way ANOVA (**A–D, H**) or two-tailed Student's t test (**F**). sWAT, subcutaneous white adipose tissue; eWAT, epididymal white adipose tissue.

The online version of this article includes the following source data and figure supplement(s) for figure 4:

**Source data 1.** Original file for the western blot analysis in **Figure 4I** (anti-pSTAT3, anti-STAT3, anti-pMAPK, anti-MAPK, anti-GAPDH, anti-UCP1, anti-β-actin).

**Source data 2.** PDF containing original scans of the relevant western blot analysis (anti-pSTAT3, anti-STAT3, anti-pMAPK, anti-MAPK, anti-GAPDH, anti-UCP1, anti-β-actin) with highlighted bands and sample labels.

**Figure supplement 1.** Interleukin-22 (IL-22) can directly act on adipocytes.

## Profiling of intestinal immune cells in mice fed NCD, HFD, or HFD-IF

To explore the mechanism by which intermittent fasting promotes the secretion of IL-22 by ILC3s, live CD45[+] lineage (CD3, CD5, B220, CD19, Gr1)[-] cells isolated from the siLP of mice fed NCD, HFD, or HFD-IF were subjected for single-cell sequencing (**Figure 6—figure supplement 1A**). Following quality controls, we analyzed 7455, 4803, 5954 single cells for the NCD, HFD, and HFD-IF groups using Seurat-V3.1, respectively. Based on singleR, we identified 25 distinct clusters of immune cells (**Figure 6A** and **Figure 6—figure supplement 1B**), including ILC1s (cluster 7 and cluster 16), ILC2s (cluster 2, cluster 12, cluster 13, and cluster 14), ILC3s (cluster 3 and cluster 5), DCs (cluster 1, cluster 4, cluster 9, cluster 10, cluster 11, cluster 17, cluster 18, and cluster 21), eosinophils (cluster 0 and cluster 6), B cells (cluster 8 and cluster 24), macrophages (cluster 15, cluster 20, and cluster 23), NKT cells (cluster 22) and mast cells (cluster 19). As expected, ILC3s expressed high RNA levels of Il7r (CD127), Rorc, and IL-22 (**Figure 6B, C** and **Figure 6—figure supplement 1C, D**). ILC3s contain two distinct cell types with highly similar expression profiles: NCR[+] and NCR[-]. These ILC3s were distinguished in our analysis as cluster 5 and cluster 3 respectively. NCR[-] ILC3s (cluster 3) were CCR6 positive and contained lymphoid tissue inducer cells (**Figure 6B**). Interestingly, NCR[-] ILC3s were characterized by high levels of vasoactive intestinal peptide receptor 2 (Vipr2) (**Figure 6C**), which is critical for the migration and function of ILC3s (**Seillet et al., 2020**; **Yu et al., 2021**; **Talbot et al., 2020**). The heatmap of cellular composition showed the differences in cell number and percentage among NCD, HFD, and HFD IF mice (**Figure 6—figure supplement 2A**). The change in ILC3 number was consistent with the flow cytometry results. The percentage of IL-22[+] cells in NCR[+] ILC3s was significantly increased by IF in mice fed HFD (**Figure 6—figure supplement 2B**). Moreover, GSEA revealed that HFD-induced decrement of cytokine-cytokine receptor interaction, in which IL-22 is included, was reversed by IF (**Figure 6—figure supplement 2C**). Besides, neuroactive ligand-receptor interaction, in which vipr2 is included, was upregulated in HFD-IF (**Figure 6—figure supplement 2**). The mRNA levels of *Vipr2* had a tendency to increase in sorted ILC3s from the small intestine of HFD-IF mice compared with that of HFD mice (**Figure 6—figure supplement 2E**). Further, gene difference analysis revealed that HFD-induced increase of Zmat4 was significantly attenuated by IF (**Figure 6D**, **Figure 6—figure supplement 2F, G**).

Further, the expression level of Hsp90ab1 in both NCR[-] ILC3s and NCR[+] was significantly increased by IF in mice fed HFD (**Figure 6D**, **Figure 6—figure supplement 2H**). Transcription factor analysis using the JASPAR database and TFBS Tools revealed that the expression of *Hsp90ab1* may be regulated by aryl hydrocarbon receptor (AHR) (**Figure 6D**). AHR is located in the Hsp90:XAP2:p23:Src chaperone protein complex in the cytoplasm without stimulation. Upon binding with the ligand, AhR translocates to the nucleus and heterodimerizes with AhR nuclear translocator to control the transcription of target genes, including AhR repressor (*Ahrr*), *Cyp1a1*, *Cyp1b1*, and *Il22* (**Stockinger et al., 2014**). Consistently, gene ontology analysis of the top 20 increased difference genes revealed that AHR signaling was one of the key pathways affected by intermittent fasting in HFD mice (**Figure 6E and F**). Furthermore, the AHR target genes, including *Il22*, *Ahrr*, *Cyp1a1*, *Cyp1b1*, increased significantly

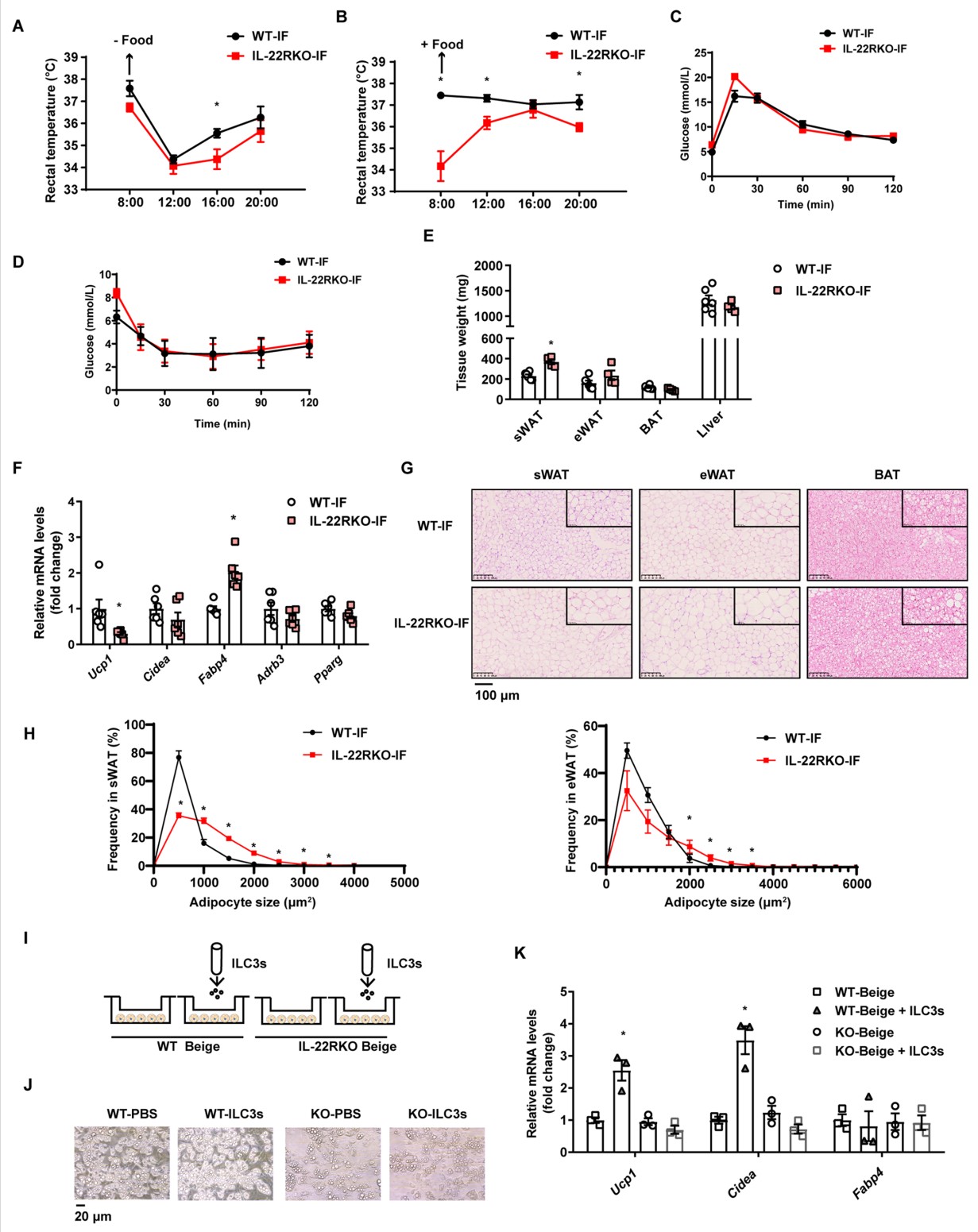

**Figure 5.** Interleukin-22R knockout (IL-22RKO) blocks white adipose tissue beigeing induced by intermittent fasting. Eight-week-old IL-22RKO and wild-type (WT) mice were subjected to alternate day fasting for 30days. n=6mice per group. (**A**) Rectal temperature of mice at room temperature during the fasting day. n=6 per group. (**B**) Rectal temperature of mice at room temperature during the fed day. n=6 per group. (**C**) Oral glucose tolerance test (OGTT) of WT-IF mice and IL-22RKO-IF mice. n=6 per group. (**D**) Insulin tolerance test (ITT) of WT-IF mice and IL-22RKO-IF mice. n=6 per group. (**E**) Tissue weight of subcutaneous white adipose tissue (sWAT), epididymal white adipose tissue (eWAT), BAT, and liver from WT-IF and IL-22RKO-IF

*Figure 5 continued on next page*

Figure 5 continued

mice. n=6mice/group. (**F**) qPCR analysis of thermogenic genes in sWAT from WT-IF and IL-22RKO-IF mice. n=6mice/group. (**G**) Representative images of hematoxylin-and-eosin-stained sections of sWAT, eWAT, and BAT (n=5 for each group). (**H**) The distribution and average adipocyte size of sWAT and eWAT were determined by ImageJ. (**I**) Schematic depicting the co-culture of type 3 innate lymphoid cells (ILC3s) with stromal vascular fraction (SVF)-induced beige adipocytes from WT or IL-22RKO mice. n=3. Experiments were repeated three times. (**J**) Phase-contrast microscopy images of beige adipocytes differentiated from SVF cells co-cultured with or without ILC3s. Shown are representatives from one experiment. (**K**) qPCR analysis of thermogenic genes in SVF-derived cells co-cultured with or without ILC3s. n=3. All data represent the means ± s.e.m. Statistical significance was determined by unpaired two-tailed Student's t test (**A–H**) or one-way ANOVA (**K**).

in ILC3s sorted from HFD-IF mice compared with HFD mice. These results indicate that intermittent fasting increases the production of IL-22 likely via activation of AhR.

## IF ameliorates the impaired interaction between intestinal myeloid cells and ILC3s

Because DCs can influence the production of IL-22 by ILC3s, we next analyzed the change in DCs induced by intermittent fasting. IF significantly attenuated the increment of inflammatory factors, such as Ccl4, Ccl17, and Ccl22, of DCs in mice fed HFD (*Figure 7A*). Kyoto Encyclopedia of Genes and Genomes (KEGG) analysis of the top 20 differentially expressed genes in cluster 17 revealed that the increase of inflammation-related pathways induced by HFD was obviously attenuated by IF (*Figure 7B and C*). These observations indicate that intermittent fasting may ameliorate the inflammatory state of the intestine by decreasing the NOD-like receptor signaling pathway in DCs (*Figure 7B and C*).

We next analyzed the intestinal cellular communication networks using CellPhoneDB analysis based on homologous gene transformation on all immune cells acquired. CellPhoneDB ligand-receptor analysis revealed hundreds of immune-to-immune interactions (*Figure 7—figure supplement 1A*). Connectome web analysis of siLP immune cells revealed strong interactions among CD8+ DCs (cluster 4), CD127+ ILC1s (cluster 16), DC8- DC-1 (cluster 9), macrophages-1 (cluster 15), CD8- DC-2 (cluster 1), macrophages-2 (cluster 23), CD8- DC-3 (cluster 10), CD8- DC-4 (cluster 11), and CD8- DC-5 (cluster 17). Notably, NCR+ ILC3s strongly interacted with CD8+ DCs (cluster 4), CD127+ ILC1 s (cluster 16), DC8- DC (cluster 9), macrophages (cluster 15), CD8- DC (cluster 1), macrophages (cluster 23), CD8- DC (cluster 10), CD8- DC (cluster 11), and CD8- DC (cluster 17). ILC subsets, DCs, and macrophages were in the central communication hubs of the healthy small intestine (*Figure 7—figure supplement 1B and C*). Analysis of highly expressed interactions uncovered various uncharacterized and validated signaling pathways implicated in intestine homeostasis in mice (*Figure 7—figure supplement 1D*). Analysis of the NCD mouse interactome suggests that macrophages (cluster 15) may interact with NCR+ ILC3s through CD74_COPA, CD74_MIF, CD44_HBEGF, CD44_FGFR2, CCL4_SLC7A1, and IL1B_ADRB2. Similarly, DCs (cluster 9) may interact with NCR+ILC3s through CD74_COPA, CD74_MIF, IL1B_ADRB2, CD44_HBEGF, and CD44_FGFR2. All these observations indicate that macrophages and DCs play critical roles in the recruitment and maintenance of ILC3s.

HFD significantly reduced the interaction between ILC subsets, DCs, and macrophages in the small intestine (*Figure 7D* and *Figure 7—figure supplement 1B*). This reduction was reversed by IF (*Figure 7E* and *Figure 7—figure supplement 1B*). Analysis of the interactomes in mice fed HFD suggested that IF significantly altered the interacting proteins (*Figure 7F and G*). To further test this observation, we sorted macrophages and ILC3s from HFD and HFD-IF mice (*Figure 7—figure supplement 2*). As shown in *Figure 7—figure supplement 2B and C*, *Ccl4* and *Cd74* increased significantly in macrophages, while the receptors expressed on ILC3s remained largely unchanged (*Figure 7—figure supplement 2*). Together, these data suggest that IF may promote the production of IL-22 from ILC3s by altering the interactome in intestinal myeloid cells and ILC3s.

## Discussion

Our present study demonstrates that intestinal ILC3s are critical for the beigeing of white adipose tissue induced by intermittent fasting. This conclusion is supported by following observations. First, intermittent fasting stimulates the secretion of IL-22 by intestinal ILC3s in either lean mice fed NCD, obese mice fed HFD, or mice with metabolic dysfunction induced by HFCD diet. Second,

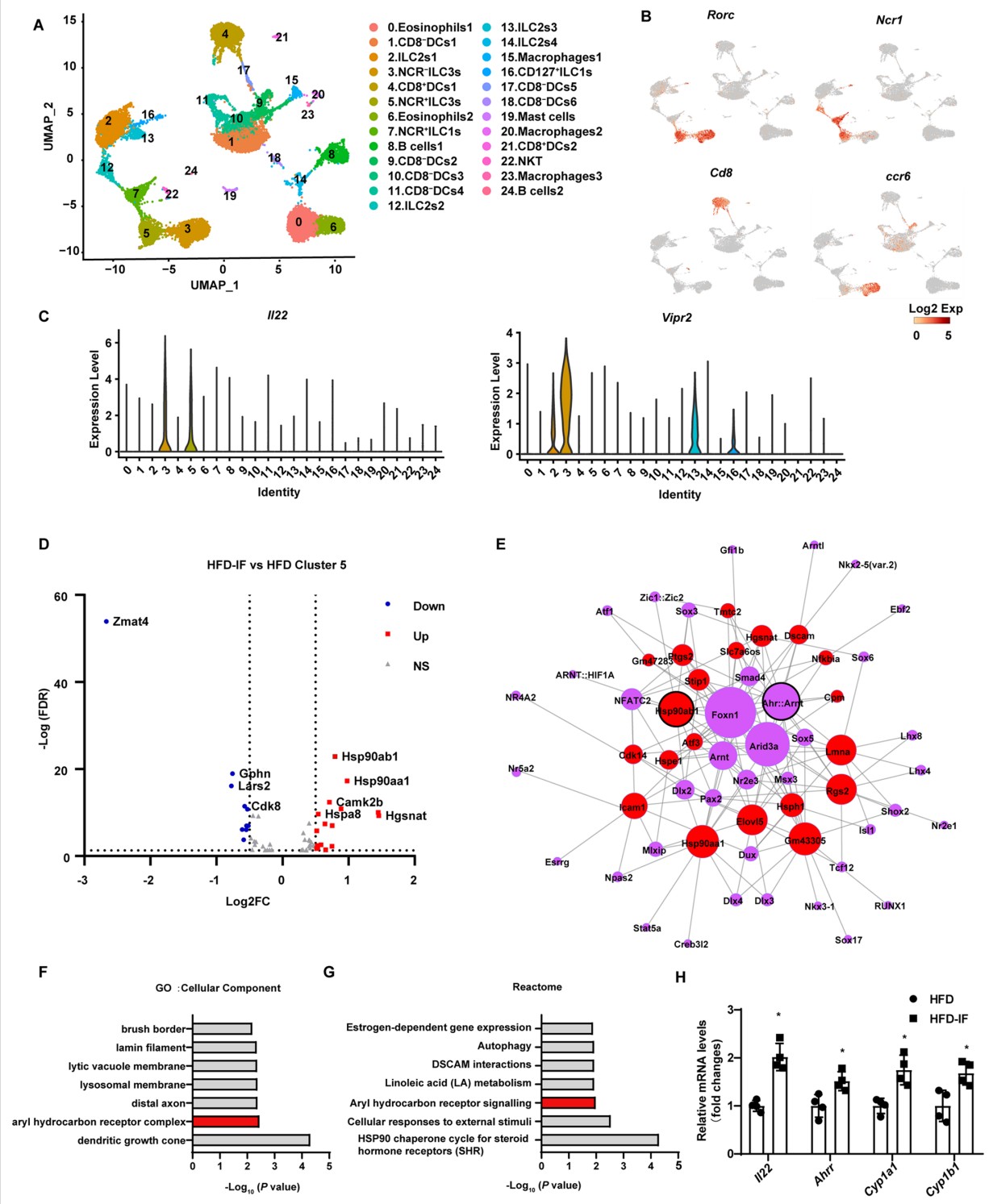

**Figure 6.** Profiling of intestinal immune cells derived from mice fed normal chow diet (NCD), high-fat diet (HFD), and high-fat diet with intermittent fasting (HFD-IF). Live CD45[+] lineage[-] cells sorted from mice fed normal chow diet (NCD group), high-fat diet (HFD group), high-fat diet with alternate day fasting (HFD-IF group) were analyzed using single-cell sequencing. (**A**) Cell subsets in the small intestine lamina propria CD45[+] lineage[-] immune cell atlas. Two-dimensional (2D) representation of cell profiles (dots) from the small intestine lamina propria, colored and numbered by cluster membership. (**B and C**) UMAP feature plots (**B**) and violin plots (**C**) showing RNA expression of cluster markers for the indicated cell populations. UMAP feature plots are based on the UMAP shown in (**A**). (**D**) The volcano plot of differentially expressed genes in cluster 5 (NCR[+] type 3 innate lymphoid cells [ILC3s]). The red dots represent upregulated genes in HFD-IF group compared with HFD group, while the blue dots represent downregulated genes in HFD-IF group

*Figure 6 continued on next page*

*Figure 6 continued*

compared with HFD group. Hsp90ab1 is one of the notably upregulated genes. (**E**) Transcription factor prediction using the Jaspar database and TFBS tools. Red dots represent differentially expressed genes, purple dots represent transcription factors, and larger nodes represent more nodes connected to them. (**F**) Gene ontology (GO) of the top 20 differentially expressed genes in cluster 5. The cellular component up GO terms in HFD-IF group compared with HFD group. (**G**) The reactome up terms of cluster 5 top 20 differential genes in HFD-IF group compared with HFD group. (**H**) qPCR analysis of AHR target genes in ILC3s sorted from HFD and HFD-IF mice. n=4. *, p<0.05. All data represent the means ±s.e.m. Statistical significance was determined by unpaired two-tailed Student's t test.

The online version of this article includes the following figure supplement(s) for figure 6:

**Figure supplement 1.** Profiling of intestinal immune cells from mice fed normal chow diet (NCD), high-fat diet (HFD), or high-fat diet with intermittent fasting (HFD-IF).

**Figure supplement 2.** Effects of intermittent fasting on the gene expression of type 3 innate lymphoid cells (ILC3s).

a comprehensive set of in vivo and in vitro experiments shows that intermittent fasting promotes adipose tissue beigeing through the intestinal ILC3-IL-22-IL-22R axis.

The role of ILC3s in metabolism remains largely unknown. Our studies provide evidence supporting the metabolic benefit of intestinal ILC3s in intermittent fasting. Adoptive transfer of intestinal ILC3s isolated from lean mice was sufficient to improve the metabolic dysfunctions in DIO mice, including increase of white adipose tissue beigeing and glucose tolerance. Interestingly, adoptive transfer of ILC3s significantly increased their number only in intestine. Further, alternate day fasting did not alter the ILC3s in adipose tissue. These observations indicate that ILC3s in intestine rather than in adipose tissue account for the metabolic benefit. In addition to directly promoting the beigeing of white adipose tissue through IL-22, intestinal ILC3s may enhance the intestinal mucosal barrier (*Ibiza et al., 2016*), reducing serum LPS and peptidoglycan, thus reducing the inhibitory effect of LPS and peptidoglycan on the beigeing of white adipose tissue (*Chen et al., 2022*). *O'Sullivan et al., 2016*, have reported that ILC3s are negligible in white adipose tissue of either lean or obese mice. Similarly, *Sasaki et al., 2019*, have demonstrated that transplanting bone marrow cells from Rag2$^{-/-}$ mice (lack of T cells, B cells) into Il2rg$^{-/-}$ Rag2$^{-/-}$ mice (lack of T cells, B cells, ILC cells) could not increase the number of ILC3s in the adipose tissue of recipient mice. However, it is worth noting that ILC3s have been detected in adipose tissue by other report. Using the marker lineage$^-$KLRG1$^-$Il-7rα$^+$Thy-1$^+$, ILC3s cells have been reported to account for approximately 20% of lineage$^-$KLRG1$^-$Il-7rα$^+$ cells in adipose tissue. In our study, we also detected ILC3s defined as CD127$^+$Lin$^-$RORγt$^+$ in adipose tissue. These ILC3s accounted for approximately 10% of CD127$^+$Lin$^-$ cells. ILC3s have also been shown to be present in human adipose tissue (*Hildreth et al., 2021*). Importantly, the proportion and density of ILC3s in white adipose tissue of people with obesity increase relevant to healthy people, and positively correlate with BMI in obese patients. Thus, the existence and function of ILC3s in mouse and human adipose tissue deserve further investigation.

Communication between intestine and metabolic organs is currently under active investigation. Our studies identify ILC3s-IL-22-IL-22R as a novel pathway mediating the crosstalk between intestine and adipose tissue. Evidence supporting this conclusion are five folds. (1) Intermittent fasting reverses the reduction of intestinal ILC3s and IL-22 in DIO mice. (2) Adoptive transfer of intestinal ILC3s enhances beigeing of white adipose tissue. (3) Co-culture of intestinal ILC3s with SVF cells increases their differentiation into beige cells. (4) Exogenous IL-22 mimics the effect of intermittent fasting on beigeing of white adipose tissue. (5) Deficiency of IL-22R blocks the IF-induced beigeing of white adipose tissue. In lines with our observation, a series of recent studies have suggested that cytokines can act directly on adipocytes to regulate thermogenesis. For example, γδ T cells regulate heat production by secreting IL-17 (*Hu et al., 2020*). Deficiency of IL-10 increases energy consumption and renders mice resistant to DIO (*Yu et al., 2021*). IL-27-IL-27Rα signaling promotes thermogenesis, prevents DIO, and improves insulin resistance (*Wang et al., 2021a*). IL-33 induces the beigeing of white adipose tissue by activating ILC2s (*Brestoff et al., 2015*). Our studies extend the effect of cytokines on thermogenesis to IL-22. IL-22 promotes heat production, renders mice resistant to reduction of body temperature induced by cold exposure, and reduces obesity induced by HFD. Together with previous report showing that IL-22 increases the lipolysis of adipocytes, our studies suggest that IL-22 can directly act on adipocytes to alter the adipose tissue homeostasis. IL-22 activates IL-22 receptor, which then regulates the expression of downstream inflammatory factors, tissue repair molecules, chemokines, antimicrobial peptides, and other molecules via Jak-1- and Tyk-2-dependent phosphorylation of STAT3. Our

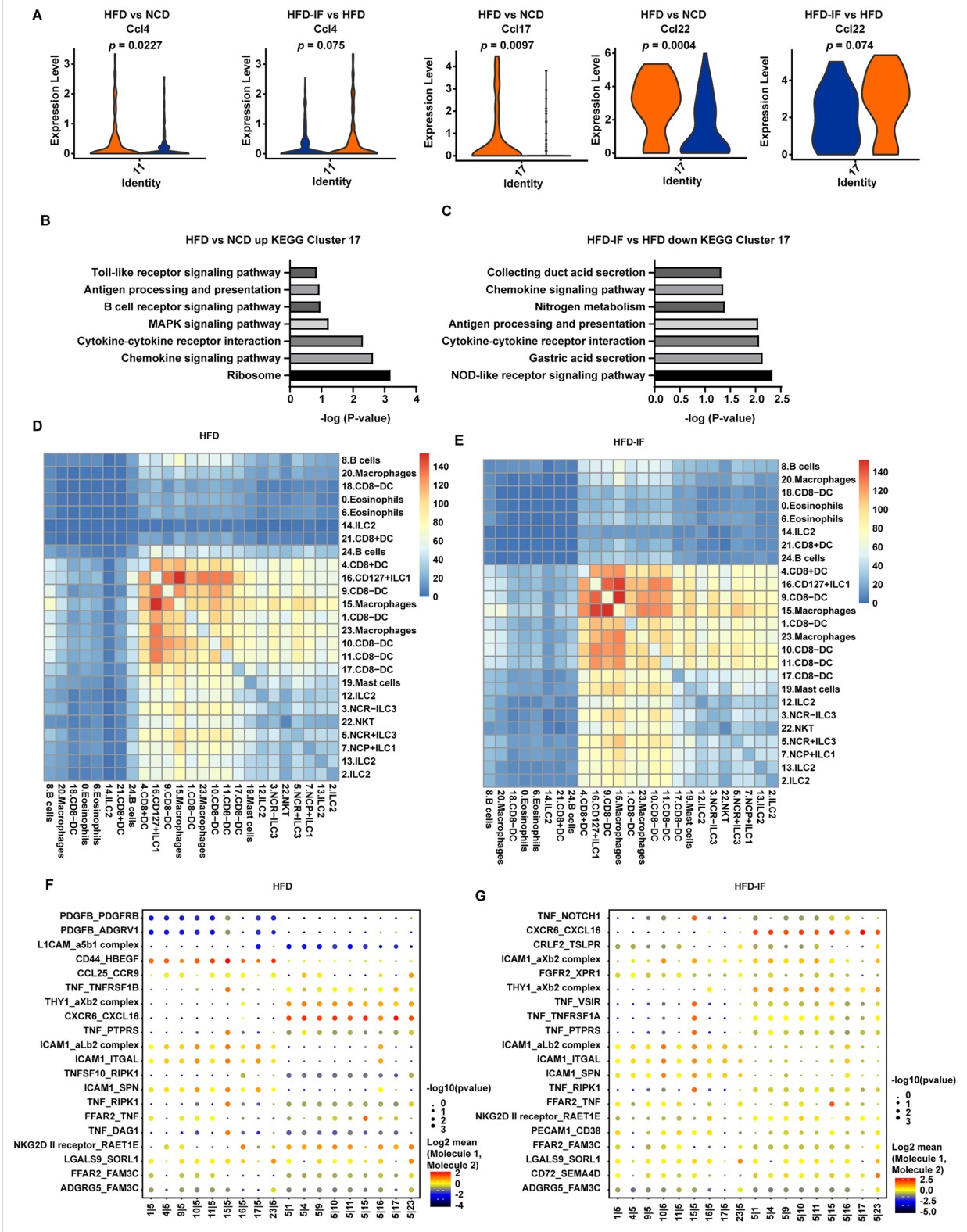

**Figure 7.** Interaction between myeloid cells and type 3 innate lymphoid cells (ILC3s). (**A**) Violin plots showing the RNA expression of chemokines in CD8⁻ dendritic cells (DCs) (cluster 11 and cluster 17). (**B**) Kyoto Encyclopedia of Genes and Genomes (KEGG) enrichment in upregulated genes of CD8⁻ DCs of high-fat diet (HFD) group compared with normal chow diet (NCD) group (cluster 17). (**C**) KEGG enrichment of downregulated genes of CD8⁻ DCs of high-fat diet with intermittent fasting (HFD-IF) group compared with HFD group (cluster 17). (**D**) Heatmap showing the number of significant interactions identified between cell types in sorted small intestine lamina propria (siLP) immune cells of HFD mice as determined by CellPhoneDB. The color

*Figure 7 continued on next page*

*Figure 7 continued*

represents the number of interactions between cell types, a higher number of interactions (red), and a lower number of interactions (blue). (**E**) Heatmap showing the number of significant interactions identified between cell types in sorted siLP immune cells of HFD-IF mice as determined by CellPhoneDB. The color represents the number of interactions between cell types: a higher number of interactions (red) and a lower number of interactions (blue). (**F**) Interaction pattern of the top 20 protein pairs and the top 20cell types in sorted siLP immune cells of HFD group mice. The x-axis is the cell-type interaction, and the y-axis is the protein interaction. The larger the point is, the smaller the p value. The color represents the average expression, and red to black indicates the level from high to low. (**G**) Interaction pattern of the top 20 protein pairs and the top 20cell types in sorted siLP immune cells of HFD-IF mice. The x-axis is the cell-type interaction, and the y-axis is the protein interaction. The larger the point is, the smaller the p value. The color represents the average expression, and red to black indicates the level from high to low.

The online version of this article includes the following figure supplement(s) for figure 7:

**Figure supplement 1.** Cell-cell interactions in the intestine.

**Figure supplement 2.** Increased expression of CD44 and CCl4 in macrophages.

studies suggest that IL-22 may promote the expression of downstream thermogenic genes in adipose tissue through IL-22R. Deficiency of IL-22R blocks the upregulation of thermogenic genes induced by intermittent fasting. SVF cells lacking IL-22R demonstrate no response to the upregulation of thermogenic genes induced by intestinal ILC3. Thus, our results provide novel evidence supporting the concept that intestinal ILC3s modulate beigeing of adipose tissue via IL-22-IL-22R pathway.

The physiological mechanism underlying the secretion of IL-22 by ILC3s remains largely unknown. Our studies reveal the distinct secretion pattern of IL-22 induced by intermittent fasting. Consistently, previous studies have shown that secretion of IL-22 by ILC3s changes between active and quiescent periods throughout the day which is regulated by the cycle patterns of food intake (*Brooks et al., 2021*). These suggest that secretion of IL-22 by ILC3s is altered by feeding rhythm. It is currently unclear how feeding influences the physiological function of ILC3s. Previous studies indicate a mechanism involving neuropeptide vasoactive intestinal peptide (VIP), which is able to stimulate the secretion of IL-22 by ILC3s (*Seillet et al., 2020*). Consistently, our single-cell RNA-seq data also showed that VIPR2 is highly expressed in intestinal ILC3s and intermittent fasting activates VIPR2 signaling pathway. Since release of VIP is stimulated by food intake, these observations indicate that VIP-VIPR signaling may coordinate with food intake to drive the production of IL-22. However, conflicting result exists. Studies by *Talbot et al., 2020* have shown that VIP inhibits ILC3 secretion of IL-22. Reasons accounting for this difference remains unknown but may be context dependent. Alternatively, we found that intermittent fasting may promote ILC3s secreting IL-22 though activating AhR signaling. In addition, feeding may decrease the levels of antimicrobial peptides secreted by epithelial cells, while increasing the expression of lipid-binding proteins (*Talbot et al., 2020*). Moreover, segmented filamentous bacteria may account for the effect of feeding on the secretion of IL-22 by intestinal ILC3s by periodically attaching to the epithelial surface (*Brooks et al., 2021*). These alterations may substantially influence the immune networks in intestine, leading to subsequent change in the secretion of IL-22 by ILC3s. In support of this concept, our single-cell RNA-seq analysis shows a significant change in the intestinal immune cell network involving DCs, macrophages, and ILC3s. Further examination should focus on dissecting the novel molecular mechanism by which intestinal DCs and macrophages interact with ILC3s and its consequence on the IL-22 production.

In conclusion, our studies suggest that intermittent fasting can promote the secretion of IL-22 by intestinal ILC3s. IL-22 promotes beigeing of white adipose tissue through IL-22R. Intestinal ILC3s thus may serve as a potential target for the intervention of metabolic disorders.

## Materials and methods

### Key resources table

| Reagent type (species) or resource | Designation | Source or reference | Identifiers | Additional information |
|---|---|---|---|---|
| Antibody | PerCP/Cy5.5 anti-mouse CD45(30-F11) (Mouse Monoclonal) | BioLegend | Cat# 103131 | FACS (1:400) |
| | *Continued on next page* | | | |

| Reagent type (species) or resource | Designation | Source or reference | Identifiers | Additional information |
|---|---|---|---|---|
| Antibody | FITC anti-mouse CD3ε(RA3-6B2) (Rabbit Monoclonal) | BioLegend | Cat# 103205 | FACS (1:400) |
| Antibody | FITC anti-mouse/human CD45R/B220(RA3-6B2) (Rabbit Monoclonal) | BioLegend | Cat# 103205 | FACS (1:400) |
| Antibody | FITC anti-mouse Ly-6G/Ly-6C(Gr-1)(RB6-8C5) (Rabbit Monoclonal) | BioLegend | Cat# 108405 | FACS (1:400) |
| Antibody | FITC anti-mouse CD19(6D5) (Rabbit Monoclonal) | BioLegend | Cat# 115506 | FACS (1:400) |
| Antibody | FITC anti-mouse CD5(53–7.3) (Rabbit Monoclonal) | BioLegend | Cat# 100605 | FACS (1:400) |
| Antibody | BV421 anti-mouse CD127(IL-7Rα) (A7R34) (Rabbit Monoclonal) | BioLegend | Cat# 135023 | FACS (1:400) |
| Antibody | PE/Cyanine7 anti-mouse CD90.2(30-H12) (Rabbit Monoclonal) | BioLegend | Cat# 105325 | FACS (1:400) |
| Antibody | PE/Cyanine7 anti-mouse CD45.1 (Mouse Monoclonal) | BioLegend | Cat# 110729 | FACS (1:400) |
| Antibody | PE anti-mouse RORγt(Q31-378) (Mouse Monoclonal) | BD Pharmingen | Cat# 562607 | FACS (1:400) |
| Antibody | Alexa Fluor 647 anti-mouse IL-22(Poly5164) (Mouse Polyclonal) | BioLegend | Cat# 516406 | FACS (1:400) |
| Antibody | BV605 anti-mouse/human KLRG1(MAFA)(2F1/KLRG1) (Syrian Hamster Monoclonal) | BioLegend | Cat# 138419 | FACS (1:400) |
| Antibody | PE anti-mouse CD117(c-kit) (2B8) (Rabbit Monoclonal) | BioLegend | Cat# 105807 | FACS (1:400) |
| Antibody | Rb polyclonal antibody to UCP1 (Rabbit Polyclonal) | abcam | Cat# ab10983 | WB (1:1000) |
| Antibody | Phospho-Stat3 (Tyr705) Rabbit mAb (Rabbit Monoclonal) | Cell Signaling Technology | Cat# 9145 | WB (1:1000) |
| Antibody | Stat3 (124H6) Mouse mAb (Mouse Monoclonal) | Cell Signaling Technology | Cat# 9139 | WB (1:1000) |
| Antibody | Phospho-p38 MAPK (Thr180/Tyr182) (D3F9) XP Rabbit mAb (Rabbit Monoclonal) | Cell Signaling Technology | Cat# 4511 | WB (1:1000) |
| Antibody | p38 MAPK (D13E1) XP Rabbit mAb (Rabbit Monoclonal) | Cell Signaling Technology | Cat# 8690 | WB (1:1000) |
| Antibody | Rb polyclonal antibody to UCP1 (Rabbit Polyclonal) | abcam | Cat# ab10983 | WB (1:1000) |
| Commercial assay or kit | eBioscience Fixation/Perm Diluent | Invitrogen | Cat# 00-8333-56 | |
| Commercial assay or kit | eBioscience Fixable Viability Dye eFluorTM 606 | Invitrogen | Cat# 65-0866-14 | FACS (1:400) |

## Animals

Four-week-old male C57BL/6 and CD45.1 mice were obtained from Charles River Laboratories (Peking, China). IL-22RKO mice (KO-00115) were purchased from BRL Medicine Inc Mice were housed in standard rodent cages and maintained in a regulated environment (21–24°C, humidity at 40–70%, 12:12 hr light:dark cycle with lights on at 8:00 AM) at the Department of Experimental Animal Science, Peking University Health Science Center. An NCD (D12450H; Research Diets) and water were available ad libitum. Obese mice (DIO) were induced with an HFD (60% fat, D12492; Research Diets) for 12 weeks. Eight-week-old C57BL/6 mice received intraperitoneal injection of saline control or IL-22 (R&D Systems, 582-ML) at a dose of 4 μg/kg/day every other day for 6 weeks. All of the animal experiments complied with the protocols for animal use, treatment and euthanasia approved by Peking University (Permit Number: LA2017099).

## Cell preparation

For isolation of siLP cells, small intestines from euthanized mice were emptied of the contents, excised of Peyer's patches, opened longitudinally and cut into 1 cm pieces. The intraepithelial lymphocytes were dissociated from the intestine fragments by first shaking the fragments for 20 min at 37°C in PBS containing 0.3% BSA, 5 mM EDTA, and 1 mM dithiothreitol. Vortex the fragments three times with

PBS containing 2 mM EDTA. To isolate lamina propria cells, the remaining fragments were minced and digested at 37°C for 50 min in RPMI 1640 medium containing 0.04 mg/mL collagenase IV (Sigma), 0.1 mg/mL deoxyribonuclease (DNase) I (Roche), and 0.5 mg/mL dispase. The digestion suspension was then filtered through a 40 μm cell strainer and centrifuged at 540×$g$ for 6 min. Cell pellets were resuspended in PBS containing 2% fetal bovine serum (FBS) for further analysis.

## Flow cytometry

Single-cell suspensions were preincubated with anti-CD16/32 (clone 2.4G2) for 10 min to block the surface Fc receptors. Then, cell-surface molecules were stained with different antibody combinations for 30 min in cell staining buffer. Dead cells were excluded with Fixable Viability Dye eFluor 606 (Invitrogen). For intracellular transcription factor staining, the cells were fixed and permeabilized with a Foxp3 staining buffer set (eBioscience) according to the manufacturer's protocol. Transcription factor staining usually lasted for more than 4 hr at 4°C. The gate strategy for ILC3s was live lineage (CD3, CD5, CD19, B220, and Gr-1)$^-$ CD127$^+$RORγt$^+$. For intracellular cytokine staining, the digested cells were first incubated in RPMI 1640 with 10% FBS and 10 ng/mL recombinant murine IL-7 (PeproTech), and stimulated with 50 ng/mL phorbol 12-myristate 13-acetate, 750 ng/mL ionomycin for 3 hr and added with 2 μM monensin for the last 2.5 hr.

Flow cytometry analyses were performed on an LSR Fortessa (BD Biosciences). The flow cytometry data were analyzed with FlowJo software (Tree Star). The antibodies used in this study are listed in Key resources table.

## ILC3s sorting, transfer, and co-culture experiment

For ILC3s transfer experiment, co-culture experiment, and single-cell RNA-seq, ILCs were collected by FACS. siLP cells from male C57BL/6J mice were prepared and stained with surface molecules for 30 min at 4°C followed by sorting on a FACS Aria III cell sorter (BD Biosciences). A gating strategy of live lineage$^-$CD127$^+$KLRG1$^-$c-Kit$^+$ was used to sort the ILC3s.

Purification checks were performed after each sort. The cells were suspended in PBS and then intravenously injected into HFD mice (200 μL PBS/mouse). ILC3s were sorted from 20 WT NCD mice, and sorting for the adoptive transfer was performed six times within 4 weeks from the first injection. The total number of ILC3 injected into HFD mice was 2.4×10$^5$ cells/mouse/experiment. Mice were placed on an HFD for 16 weeks before ILC3s were transferred. In addition, to confirm that these cells indeed originate from NCD mice, we transferred ILC3s from CD45.1 mice to WT mice and detected the CD45.1 positive cells in recipient mice.

## Single-cell RNA-seq

Single cells were captured via the GemCode Single Cell Platform using the GemCode Gel Bead, Chip, and Library Kits (10x Genomics) according to the manufacturer's protocol. Briefly, flow-sorted cells were suspended in PBS containing 0.4% BSA and loaded at 6000 cells per channel. The cells were then partitioned into a GemCode instrument, where individual cells were lysed and mixed with beads carrying unique barcodes in individual oil droplets. The products were subjected to reverse transcription, emulsion breaking, cDNA amplification, shearing, 5′ adaptor, and sample index attachment. Libraries were sequenced on a HiSeq 2500 (Illumina).

## Isolation of SVF cells and induction of beigeing

Isolation of SVF and induction of beigeing were performed as reported (*Aune et al., 2013*). Briefly, sWAT from male C57BL/6 mice was digested with 1 mg/mL type 2 collagenase (Sigma-Aldrich) at 37°C for 40 min with shaking. Digestion was terminated by complete DMEM/F12 medium containing 10% FBS, 100 units/mL penicillin, and 100 units/mL streptomycin (Invitrogen, CA, USA). The cell suspension was centrifuged at 700×$g$ for 10 min to separate floating adipocytes from the SVF pellets. The pellets containing the SVF cells were resuspended in complete medium and filtered using a 70 μm diameter filter. The cell suspension was then centrifuged at 700×$g$ for 10 min. Cell pellets were resuspended and mixed well and then plated on dishes in complete medium. Cells were grown to 95% confluence in complete medium and then differentiated into beige or white adipocytes as previously described (*Seale et al., 2011*). Briefly, for induction to beige adipocytes, cells were cultured for 2 days with induction medium supplemented with 5 μg/mL insulin, 1 nmol/L T3, 1 μmol/L rosiglitazone,

125 µmol/L indomethacin, 0.5 mmol/L isobutylmethylxanthine, and 5 µmol/L dexamethasone. Cells were then cultured in maintenance medium supplemented with 5 µg/mL insulin, 1 nmol/L T3, and 1 µmol/L rosiglitazone for 4 days. Fresh media were replaced every 2 days.

## Co-culture experiments

To determine the direct effect of ILC3s on beige adipocytes differentiation, SVFs were co-cultured with ILC3s using a transwell system (0.4 µm pore size, BD Biosciences). In brief, SVFs isolated from adipose tissue were grown in the bottom chamber of the transwell insert in a 12-well plate and induced into beige adipocytes, while sorted ILC3s were seeded in the upper chamber. SVFs cultured alone and SVF cells co-cultured with CD127⁻ cells were used as controls. After co-culture for 72 hr, beige adipocytes in the lower chamber were collected for further detection.

## RNA isolation and qPCR analysis

RNA was extracted from cells or adipose tissue using RNATrip (Applied Gene, Beijing, China) and reverse-transcribed into cDNAs with Hifair III 1st Strand cDNA Synthesis SuperMix (Yeasen). SYBR Green-based quantitative real-time PCR was performed using the Agilent Aria Mx real-time PCR system. The primer sequences used in this study are listed in *Supplementary file 1*.

## Western blot analysis

Differentiated SVF cells and sWAT were homogenized using RIPA lysis buffer. Proteins were separated by 10% sodium dodecyl sulfate-polyacrylamide gel electrophoresis and then transferred onto a nitro-cellulose membrane. Membranes were incubated in 5% fat-free milk for 1 hr at room temperature and then incubated with primary antibodies overnight at 4°C. The reaction was detected with IRDye-conjugated secondary antibody and visualized using the Odyssey infrared imaging system (LI-COR Biosciences).

## Histological studies

Paraffin-embedded sWAT sections were stained with hematoxylin-eosin (H&E). Images were scanned using a NanoZoomer-SQ (Hamamatsu).

## Oil red O staining

Cells were washed with PBS for three times, and then fixed with 4% paraformaldehyde for 30 min. Next, the cells were stained with oil red O for 60 min, and washed with PBS for 5 min. Cells were then stained with hematoxylin for 5 s and washed with PBS. The dyed cells were photographed under the microscope (Leica, Germany).

## Enzyme-linked immunosorbent assay

Levels of IL-22 were measured by double-antibody sandwich ELISAs (M2200; R&D Systems). Briefly, 100 µL of Assay Diluent were added to each well. 50 µL of blood samples and standards were added and incubated at room temperature for 2 hr. After washing, conjugates were added and incubated for 2 hr. After washing, the substrate solution was added and incubated for 30 min and washed. And stop solution were added finally. Optical density values were measured using a microplate reader (Bio-Rad, Hercules, CA, USA).

## OGTT and ITT

Oral glucose tolerance tests (OGTT) and insulin tolerance tests (ITT) were performed after 16 or 6 hr of fasting, respectively. Mice were given with 3 g/kg glucose by gavage or injected intraperitoneally with 0.75 U/kg insulin. Blood samples were collected from the tail vain at 0, 15, 30, 60, 90, and 120 min after glucose or insulin treatment. Blood glucose levels were measured using a glucometer (Roche, Basel, CH).

## Cold exposure

Mice were placed in a 4°C cold room for 6 hr. The rectal temperature was measured every hour during the cold challenge with a rectal probe (Braintree Scientific, Braintree, MA, USA).

## Statistical analysis

Data were analyzed by GraphPad Prism software v.8.0 and presented as the mean ± s.e.m. The Shapiro-Wilk normality test was used to determine the normal distribution of samples. Unpaired Student's t test (normal distribution) or Mann-Whitney U tests (non-normal distribution) were used to analyze data between two groups and one-way ANOVA followed by Bonferroni's multiple-comparisons test (normal distribution) or Kruskal-Wallis test (non-normal distribution) was used for three or more groups. The sample sizes were determined by power analysis using StatMate v.2.0. No data were excluded during the data analysis.

## Acknowledgements

This research was supported by grants from the National Natural Science Foundation of China (81930015, 81730020, 82070592, and 82270610), National Institutes of Health Grant R01DK112755 and 1R01DK129360.

## Additional information

### Funding

| Funder | Grant reference number | Author |
|---|---|---|
| National Natural Science Foundation of China | 81930015 | Weizhen Zhang |
| National Natural Science Foundation of China | 81730020 | Weizhen Zhang |
| National Natural Science Foundation of China | 82070592 | Yue Yin |
| National Natural Science Foundation of China | 82270610 | Yue Yin |
| National Institutes of Health | R01DK112755 | Weizhen Zhang |
| National Institutes of Health | 1R01DK129360 | Weizhen Zhang |

The funders had no role in study design, data collection and interpretation, or the decision to submit the work for publication.

### Author contributions

Hong Chen, Conceptualization, Resources, Data curation, Software, Formal analysis, Validation, Investigation, Methodology, Writing - original draft, Project administration, Writing – review and editing; Lijun Sun, Resources, Data curation, Validation, Methodology, Project administration; Lu Feng, Data curation, Formal analysis, Investigation; Xue Han, Formal analysis; Yunhua Zhang, Resources, Data curation, Project administration; Wenbo Zhai, Methodology, Project administration; Zehe Zhang, Project administration; Michael Mulholland, Resources, Visualization; Weizhen Zhang, Conceptualization, Resources, Formal analysis, Supervision, Funding acquisition, Investigation, Visualization, Methodology, Writing – review and editing; Yue Yin, Conceptualization, Resources, Data curation, Formal analysis, Supervision, Funding acquisition, Investigation, Writing – review and editing

### Author ORCIDs

Weizhen Zhang [ID] http://orcid.org/0000-0001-8791-2798
Yue Yin [ID] https://orcid.org/0000-0001-6497-5382

### Ethics

All of the animal experiments complied with the protocols for animal use, treatment and euthanasia approved by Peking University (Permit Number: LA2017099).

Reviewer #1 (Public Review): https://doi.org/10.7554/eLife.91060.3.sa1
Reviewer #2 (Public Review): https://doi.org/10.7554/eLife.91060.3.sa2
Reviewer #3 (Public Review): https://doi.org/10.7554/eLife.91060.3.sa3
Author Response https://doi.org/10.7554/eLife.91060.3.sa4

## Additional files

### Supplementary files

• Supplementary file 1. Sequences of primers used in quantitative PCR.

• MDAR checklist

### Data availability

All of the data supporting the findings of this study are included in the article and supplementary information. The Single-cell RNA sequencing data were uploaded to a public database (PRJNA1031585).

The following dataset was generated:

| Author(s) | Year | Dataset title | Dataset URL | Database and Identifier |
|---|---|---|---|---|
| Hong C | 2023 | ScRNAseq provides new information into intestine-resident immune cell profiling in response to repeated fasting and refeeding | https://www.ncbi.nlm.nih.gov/bioproject/PRJNA1031585 | NCBI BioProject, PRJNA1031585 |

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
