## [Editor Report · eLife assessment]

This study provides **valuable** findings showing the production of IL-22 from intestinal ILC3 during intermittent fasting promotes beigeing of white adipose tissue. The authors provided **solid** data and mechanistic insight by which IL-22-derived from ILC3 directly induces beigeing.

---

## [Referee Report · Reviewer #1 (public Review)]

In the present study, the authors carefully evaluated the metabolic effects of intermittent fasting on normal chow and HFD fed mice and reported that intermittent fasting induces beiging of subcutaneous white adipose tissue. By employing complementary mouse models, the authors provided compelling evidence to support a mechanism through ILC3/IL-22/IL22R pathway. They further performed comprehensive single-cell sequencing analyses of intestinal immune cells from lean, obese, obese undergone intermittent fasting mice and revealed altered interactome in intestinal myeloid cells and ILC3s by intermittent fasting via activating AhR. Overall, this is a very interesting and timely study uncovering a novel connection between intestine and adipose tissue in the context of executing metabolic benefits of intermittent fasting.

(1) The authors showed increased plasma IL-22 and its expression in intestine. Are intestinal ILC3s the main source of plasma IL-22?

(2) The authors transplanted intestinal ILC3s from NCD mice to DIO mice and showed significant metabolic improvements. However, in Fig. 1, intermittent fasting increased IL-22-positive ILC3s proportion rather than changing the total number. Please clarify whether this transplantation is due to increasing ILC3s number or introducing more IL-22 positive ILC3s (which are decreased in DIO). Are these transplanted ILC3s by default homing to intestine rather than to other tissues?

(3) The authors adopted cold challenge at 4 degree for 6 hours to assess beiging in subcutaneous WAT and showed difference in core temperature. However, thermogenesis in this acute cold challenge is mainly by brown adipose tissue. Beiging is a chronic and adaptive response. Based on the data in WAT, there is a beiging phenotype, but the core body temperature in acute cold challenge is not an accurate readout. It would be a missed opportunity by not evaluating thermogenic activity in BAT.

More browning genes should be included to strengthen the beiging phenotype of WAT. Moreover, inflammation in WAT can be examined to provide a whole picture of adipose tissue remodeling through this pathway.

(4) For the SVF beige adipocyte differentiation, 100 ng/mL IL-22 was used. This is highly above the physiological concentration at ~5 pg/mL. Please justify this high concentration used.

The authors showed increased Ucp1 and Cidea expression by IL-22 treatment in SVFs. Please be aware that these increases are likely due to boosted adipogenesis as told by the morphology. Please examine more adipogenic markers to confirm. Is this higher adipogenesis caused by the high concentration of IL-22?

In line 201, the authors drew the conclusion that IL-22 increased SVF beige differentiation. To fully support this conclusion, the authors should assure adipogenesis at the same baseline and then compare beiging, or examine the effect of IL-22 on normal adipogenesis to compare with beige differentiation.

---

## [Referee Report · Reviewer #2 (Public review)]

Summary:

This study aims to investigate the mediatory role of intestinal ILC3-derived IL-22 in intermittent fasting-elicited metabolic benefits.

Strengths:

The observation of induction of IL-22 production by intestinal ILC3 is significant, and the scRNAseq provides new information into intestine-resident immune cell profiling in response to repeated fasting and refeeding.

Weaknesses:

The experimental design for some studies needs to be improved to enhance the rigor of overall study. There is a lack of direct evidence showing that the metabolically beneficial effects of IF are mediated by intestinal ILC3 and their derived IL-22. The mechanism by which IL-22 induces thermogenic program is unknown. The browning effect induced by IF may involve constitutive activation of lipolysis, which was not considered.

Majority of weaknesses have been addressed in the revision. Based on the analysis of thermogenic genes in addition to Ucp1 (Fig. 4D and S6F), the alteration on thermogenesis induced by IL-22 is dependent on UCP1 but not other markers such as PGC1a, PPARg, and Cidea. The data need to be discussed in the Section of Discussion.

---

## [Referee Report · Reviewer #3 (Public review)]

Chen et al. investigated how intermittent fasting causes metabolic benefits in obese mice and find that intestinal ILC3 and IL-22-IL-22R signaling contribute to the beiging of white adipose tissue (WAT) and consequent metabolic benefits including improved glucose and lipid metabolism in diet-induced obese mice. They demonstrate that intermittent fasting causes increased IL22+ILC3 in small intestines of mice. Adoptive transfer of purified intestinal ILC3 or administration of exogenous IL-22 can lead to increases in UCP1 gene expression and energy expenditure as well as improved glucose metabolism. Importantly, the above metabolic benefits caused by intermittent fasting are abolished in IL-22R-/- mice. Using an in vitro experiment, the authors show that ILC3-derived IL-22 may directly act on adipocytes to promote SVF beige differentiation. Finally, by performing sc-RNA-seq analysis of intestinal immune cells from mice with different treatments, the authors indicate a possible way of intestinal ILC3 being activated by intermittent fasting. Overall, this study provides a new mechanistic explanation for the metabolic benefits of intermittent fasting and reveals the role of intestinal ILC3 in the enhancement of the whole-body energy expenditure and glucose metabolism likely via IL-22-induced beige adipogenesis.

Although this study presents some interesting findings, particularly IL-22 derived from intestinal ILC3 could induce beiging of WAT by directly acting on adipocytes, the experimental data are not sufficient to support the key claims in the manuscript.

---

## [Author Response]

The following is the authors’ response to the original reviews.

Comment 1: The authors showed increased plasma IL-22 and its expression in the intestine. Are intestinal ILC3s the main source of plasma IL-22?

Reply: ILC3s are the main source of IL-22 as reported previously (PMID: 30700914). In the small intestine, ILC3s account for about 62% of IL22+ cells. Other IL22+ cells include γδ T, Foxp3+T and CD4+T cells.

Comment 2: The authors transplanted intestinal ILC3s from NCD mice to DIO mice and showed significant metabolic improvements. However, in Fig. 1, intermittent fasting increased IL-22positive ILC3s proportion rather than changing the total number. Please clarify whether this transplantation is due to increasing ILC3s number or introducing more IL-22 positive ILC3s (which are decreased in DIO). Are these transplanted ILC3s by default homing to the intestine rather than to other tissues?

Reply: We believe that the transplantation increases ILC3s number, leading to the increment in IL22 levels. The transplanted ILC3s by default are homing to the intestine rather than to other tissues because ILC3s express several homing receptors such as CCR7, CCR9, and α4β7, which modulate their capacity to migrate to the gut (PMID: 26141583; PMID: 26708278; PMID: 25575242; PMID: 34625492). Our observation that ILC3s in adipose tissue remained unchanged by ILC3 cell transplantation (Supplementary Figure 5F) also supports this concept.

Comment 3: Thermogenesis in this acute cold challenge is mainly by brown adipose tissue. Beiging is a chronic and adaptive response. Based on the data in WAT, there is a beiging phenotype, but the core body temperature in acute cold challenge is not an accurate readout. It would be a missed opportunity by not evaluating thermogenic activity in BAT. More browning genes should be included to strengthen the beiging phenotype of WAT. Moreover, inflammation in WAT can be examined to provide a whole picture of adipose tissue remodeling through this pathway.

Reply: Per suggestion, we performed additional experiments to measure levels of inflammation genes such as Il4, Il1b, Il6, Il22, Il23, Il17a. As shown in supplemental figure 2D, these inflammation relevant genes were not altered.

Comment 4: For the SVF beige adipocyte differentiation, 100 ng/mL IL-22 was used. This is highly above the physiological concentration at ~5 pg/mL. Please justify this high concentration used.

Reply: We agree with the reviewer that the dose of IL-22 used is high. However, the efficient dose at 100 ng/ml used in our studies is consistent with the literatures. Previous reports have shown that IL-22 directly activates Stat3 in adipose tissue and primary adipocytes, and promotes the expression of genes involved in triglyceride lipolysis (Lipe and Pnpla2) and fatty-acid β-oxidation (Acox1) at the dose of 100 ng/ml (Wang X, Ota N, et al. Nature. 2014). Consistently, other studies have reported that IL-22 at 100 ng/ml significantly reversed the enhanced expression of CCL2, CCL20 and IL1B mRNAs in granulosa cells in vitro (Qi X, et al. Nat Med. 2019).

Comment 5: The authors showed increased Ucp1 and Cidea expression by IL-22 treatment in SVFs. Please be aware that these increases are likely due to boosted adipogenesis as told by the morphology. Please examine more adipogenic markers to confirm. Is this higher adipogenesis caused by the high concentration of IL-22?

Reply: Per suggestion, we examined the expression of adipogenic marker genes such as Pparγand Fabp4. We found that IL-22 did not increase the levels of these adipogenic marker genes relevant to the PBS control as shown in supplemental figure 6F.

**Author response image 1. sa4fig1:** 

Comment 6: In line 201, the authors drew the conclusion that IL-22 increased SVF beige differentiation. To fully support this conclusion, the authors should assure adipogenesis at the same baseline and then compare beiging, or examine the effect of IL-22 on normal adipogenesis to compare with beige differentiation.

Reply: We examined the expression of adipogenic marker genes such as Pparγ and Fabp4 and found that IL-22 did not increase the expression of these adipogenic marker genes relevant to the PBS control.

**Reviewer #2:**
This study aims to investigate the mediatory role of intestinal ILC3-derived IL-22 in intermittent fasting-elicited metabolic benefits.Strengths:The observation of induction of IL-22 production by intestinal ILC3 is significant, and the scRNAseq provides new information into intestine-resident immune cell profiling in response to repeated fasting and refeeding.Weaknesses:The experimental design for some studies needs to be improved to enhance the rigor of the overall study. There is a lack of direct evidence showing that the metabolically beneficial effects of IF are mediated by intestinal ILC3 and their derived IL-22. The mechanism by which IL-22 induces a thermogenic program is unknown. The browning effect induced by IF may involve constitutive activation of lipolysis, which was not considered.Comment 1: Lack of direct evidence showing that IL-22-expressing ILC3s in intestine is the key contributor to intermittent fasting (IF)-mediated elevation of circulating IL-22 levels. The fraction of IL-22-expressing cells was increased threefold by IF but the increase in circulating IL-22 is moderate (Figs. 1J and 1K).

Reply: IL-22 in circulation is subjected to clearance, degradation, and binding with plasma proteins, et al. Thus, circulating levels of IL-22 may be much lower than the amount secreted by the intestinal IL-22 positive ILC3s.

Comment 2: The loss of fat mass by IF suggests that the active lipolysis may explain the white fat browning which was not considered. This may apply to the observations in IL-22 treated mice as well as IL-22R KO mice.

Reply: We analyzed the expression of genes relate to lipolysis in NCD and NCD-IF mice and found that IF did not alter the levels of these genes in white adipose tissues (Supplementary figure 2D). We have addressed this concerns in lines 119, page 6.

**Author response image 2. sa4fig2:** 

Comment 3: IL-22 administration and adoptive transfer of ILC3 had no significant effect on body weight. Not clear how IL-22 improves insulin sensitivity in this case.

Reply: Our results are consistent with previous report showing that IL-22 administration improves insulin sensitivity without change in body weight (Qi X, et al. Nat Med. 2019). In addition, previous studies have demonstrated that IL-22 can increase Akt phosphorylation in muscle, liver and adipose tissues, leading to improvement in insulin sensitivity (Wang X, et al. Nature. 2014). We have addressed this potential mechanism in lines192-195, page 9.

Comment 4: The energy expenditure data look unusual given that there was little increase in oxygen consumption during dark cycle compared to light cycle (Fig.3).

Reply: The not so obvious difference in oxygen consumption between dark cycle and light cycle may be due to the technical problem of the system.

Comment 5: The thermogenic capacity for the whole fat pad needs to consider the expression of UCP1 in certain amount of tissue and the total mass for each individual animal because the mRNA level itself does not reflect the whole tissue capacity.

Reply: We used the whole subcutaneous adipose tissue from one side for qPCR to reflect the whole tissue capacity.

Comment 6: The design of studies for the adoptive transfer of ILC3 was concerned. The PBS is not a good control for the group with ILC3 cells (Figs. 2A-2H). Similar issue applies for the co-culture study in which beige only is not an ideal control for Beige+ILC3 (Figs. 2I-2J).

Reply: We agree with the reviewer that the PBS is not a good control. Because we cannot find a similar immune cell without any effect on adipocytes, we designed this experiment based on other studies in which saline or PBS are used as ILC transfer experiment controls (Sasaki T, et al. Cell Rep. 2019; Wang H, et al. Nat Commun. 2019)

Comment 7: The induction of thermogenesis by IL-22 treatment may be related to enhanced differentiation rather than direct activation of thermogenic genes (Figs. 4G and 4H).

Reply: Our observation that IL-22 did not alter the levels of genes related to adipogenesis (Supplemental figure 6F) indicates that IL-22 may not alter the differentiation of adipocytes. We addressed this concern in Lines 211-212, page 10.

**Reviewer #3:**
Chen et al. investigated how intermittent fasting causes metabolic benefits in obese mice and found that intestinal ILC3 and IL-22-IL-22R signaling contribute to the beiging of white adipose tissue (WAT) and consequent metabolic benefits including improved glucose and lipid metabolism in diet-induced obese mice. They demonstrate that intermittent fasting causes increased IL22+ILC3 in small intestines of mice. Adoptive transfer of purified intestinal ILC3 or administration of exogenous IL-22 can lead to increases in UCP1 gene expression and energy expenditure as well as improved glucose metabolism. Importantly, the above metabolic benefits caused by intermittent fasting are abolished in IL-22R-/- mice. Using an in vitro experiment, the authors show that ILC3derived IL-22 may directly act on adipocytes to promote SVF beige differentiation. Finally, by performing sc-RNA-seq analysis of intestinal immune cells from mice with different treatments, the authors indicate a possible way of intestinal ILC3 being activated by intermittent fasting. Overall, this study provides a new mechanistic explanation for the metabolic benefits of intermittent fasting and reveals the role of intestinal ILC3 in the enhancement of the whole-body energy expenditure and glucose metabolism likely via IL-22-induced beige adipogenesis.Although this study presents some interesting findings, particularly IL-22 derived from intestinal ILC3 could induce beiging of WAT by directly acting on adipocytes, the experimental data are not sufficient to support the key claims in the manuscript.Comment 1: Only increased UCP1 expression on mRNA level is not enough to support the beiging of WAT. More methods such as western blotting and immunostaining of UCP1 in WAT are needed to confirm the enhanced beige adipogenesis.

Reply: Additional experiments have been performed to measure the UCP1 protein by Western blot. The data is included in Figure 4I and Supplementary Figure 2E.

Comment 2: IL-22 is known to modulate metabolic pathways via multiple downstream functions. The use of whole-body knockout of IL-22R could not exclude the indirect effect on the promotion of beiging of WAT. Specific deletion of IL-22R in adipose tissues is therefore needed to confirm the direct effect of IL-22 on adipocytes which is suggested by the in vitro study.

Reply: We agreed with the reviewer that specific deletion of IL-22R in adipose tissues is critical to confirm the direct effect of IL-22 on adipocytes. We will generate the AdioQ-IL-22R-/- mice to test this concept further in vivo.

Comment 3: The authors failed to show the cellular distribution of IL-22R in adipose tissues. This is important because the mechanism that explains the increased beige adipogenesis could be different based on the expression of IL-22R in adipose progenitor cells or mature adipocytes. So it is not appropriate to conclude that "IL-22 then directly activates IL-22R on adipocytes, leading to subsequent induction of beiging of white adipose tissue" in line 407. Additionally, Oil red O staining is needed for Fig 4G and Fig 5J, and protein levels of UCP1 and adipogenesis-related markers are needed to evaluate beige fat differentiation and the whole adipogenesis.

Reply: Per suggestion, we have added the expression of IL-22R in adipose progenitor cells or mature adipocytes (Supplementary Figure 6E). In addition, protein levels of UCP1 and adipogenesis-related markers to evaluate the whole adipogenesis (Figure 4I, Supplementary figure 6F) are now included. We have also addressed this issue in lines 207-215, page 10.

Comment 4: Although the authors provided some hypothesis about how intermittent fasting increases IL-22+ILC3 in small intestines by sc-RNA-seq analysis, some functional assays are needed to identify the factors, for example, how about the levels of macrophage-derived IL-23 or AHR ligands in small intestines and whether they contribute to increased percentages of intestinal IL-22+ILC3 following intermittent fasting.

Reply: We used flow cytometry sorting of macrophages combined with qPCR experiments to preliminarily demonstrate that intermittent fasting increases the expression of molecules such as Cd44 and CCl4 (Supplementary Figure 10B), which may contribute to the increase in the proportion of IL-22+ ILC3s in the intestine under intermittent fasting. Our observation that IL-23 mRNA levels were not changed indicates that this molecule may not the major contributor for the communication between macrophage and ILC3s. Other potential molecules such as AHR ligands remain to be explored.

Comment 5: What are the differences between adipose ILC3 and intestinal ILC3? Why do transferred ILC3 only migrate to the small intestine but not WAT of recipient mice? It would be better to examine or at least discuss whether other factors from intestinal ILC3 may also contribute to beiging of WAT following intermittent fasting.

Reply: Intestinal ILC3s specifically express gut homing receptors CCR7, CCR9, and α4β7 (PMID: 26141583; PMID: 26708278; PMID: 25575242; PMID: 34625492). This may explain transplantation of intestinal ILC3s can migrate mainly to the intestine instead of adipose tissue (PMID: 34625492). The proportion of ILC3s in adipose tissue of mice is very small. Their functions have not been clarified yet. We have addressed this issue in lines 156-158, page 8.

There are some other factors from intestinal ILC3 which may also contribute to beiging of WAT following intermittent fasting. By secreting IL-22, ILC3 enhanced the intestinal mucosal barrier, leading to reduction of the influx of LPS and PGN into the bloodstream under high-fat diet conditions, and subsequent increase in the beiging of white adipose tissue (Chen H, et al. Acta Pharm Sin B. 2022). We have addressed this potential mechanism in lines 344-347, page 16.

Comment 6: The sensitivity of the IL-22 ELISA kit used in the manuscript was 8.2 pg/mL, according to the information from the methods, however, in Fig. 1J and Fig. 2B, the IL-22 levels in mouse plasma were lower than 6 pg/mL, which was below the sensitivity of the ELISA kit and also the assay range. Please explain.

Reply: We have double-checked the original data and found that we have made a mistake in calculating the concentration of IL-22. We have corrected this error (Fig. 1J, Fig. 2B).

Comment 7: In Fig 7A, the significance of the Hypothesis testing should be marked. In Fig 7F and 7G, the contrast between the two groups is not apparent, other comparing ways could be used to enhance the readability.

Reply: Per suggestion, we have marked the significance of the hypothesis testing between HFD vs NCD and HFD-IF vs HFD in Fig7A. Shown in Fig 7F and 7G are the top 20 enriched interacting proteins between different cell types. The dot plot displays the average expression level and significance of protein interactions in cell types.

Comment 8: The total food intake of fasting mice fed with NCD or HFD was less than those without fasting, and the food intake rate the author showed in Fig S1 represents the value that was normalized to body weight. So the author should describe it precisely In line 114.

Reply: We have revised the statement accordingly in line 114-115.

Comment 9: Western blotting analysis has been described in methods, however, there is no corresponding experimental data in the result part.

Reply: The Western blotting results are now included.